# Reasonable Effectiveness of Random Weighting: A Litmus Test for Multi-Task Learning

**Baijiong Lin**[1]                                          *bj.lin.email@gmail.com*

**Feiyang Ye**[1,2,3]                                        *12060007@mail.sustech.edu.cn*

**Yu Zhang**[1,4,*]                                          *yu.zhang.ust@gmail.com*

**Ivor W. Tsang**[3,2]                                       *ivor.tsang@gmail.com*

[1] *Department of Computer Science and Engineering, Southern University of Science and Technology*
[2] *Australian Artificial Intelligence Institute, University of Technology Sydney*
[3] *Centre for Frontier AI Research, A\*STAR*
[4] *Peng Cheng Laboratory*

**Reviewed on OpenReview:** *https://openreview.net/forum?id=jjtFD8A1Wx*

## Abstract

Multi-Task Learning (MTL) has achieved success in various fields. However, training with equal weights for all tasks may cause unsatisfactory performance for part of tasks. To address this problem, there are many works to carefully design dynamical loss/gradient weighting strategies but the basic random experiments are ignored to examine their effectiveness. In this paper, we propose the Random Weighting (RW) methods, including Random Loss Weighting (RLW) and Random Gradient Weighting (RGW), where an MTL model is trained with random loss/gradient weights sampled from a distribution. To show the effectiveness and necessity of RW methods, theoretically, we analyze the convergence of RW and reveal that RW has a higher probability to escape local minima, resulting in better generalization ability. Empirically, we extensively evaluate the proposed RW methods to compare with twelve state-of-the-art methods on five image datasets and two multilingual problems from the XTREME benchmark to show that RW methods can achieve comparable performance with state-of-the-art baselines. Therefore, we think the RW methods are important baselines for MTL and should attract more attention.

## 1 Introduction

Multi-Task Learning (MTL) (Caruana, 1997; Zhang & Yang, 2021; Vandenhende et al., 2021) aims to jointly train several related tasks to improve their generalization performance by leveraging common knowledge among them. Since MTL could not only significantly reduce the model size as well as speed up the inference but also improve the performance, it has been successfully applied to various fields (Zhang & Yang, 2021). However, when all the tasks are not highly related, which may be reflected via conflicting gradients or dominating gradients (Yu et al., 2020), it is more difficult to train an MTL model than training them separately because some tasks dominantly influence model parameters, leading to unsatisfactory performance for other tasks. This phenomenon is referred to as the *task balancing* problem (Vandenhende et al., 2021) in MTL.

Recently, several works focus on tackling this issue from an optimization perspective via dynamically weighting task losses or balancing task gradients in the training process, called *loss balancing* and *gradient balancing*

---

*Corresponding author.

methods, respectively. However, all of the existing works take Equal Weighting (**EW**) which uses the fixed and equal weights in the whole training process as a basic baseline to test the effectiveness of their methods. We think that this baseline is not sufficient and it is quite necessary to conduct a random experiment where weights are sampled randomly as a baseline to show the effectiveness of those well-designed dynamically weighting strategies, but it is missing in the existing literature.

Therefore, in this paper, we propose the Random Weighting (**RW**) methods including Random Loss Weighting (**RLW**) and Random Gradient Weighting (**RGW**) as more reasonable baselines to test loss and gradient balancing methods, respectively. Specifically, in each training iteration, we first sample loss/gradient weights from a distribution with some possible normalization and then minimize the aggregated loss/gradient weighted by the random loss/gradient weights. Although the RW methods seem unreasonable, they can not only converge but also achieve comparable performance with existing methods that use carefully tuned weights. Thus, we think the RW methods are important baselines for MTL and deserve more attention.

To better understand the effectiveness and necessity of RW methods, we provide both theoretical analyses and empirical evaluations. Theoretically, we show RW methods are the stochastic variants of EW. From this perspective, we give a convergence analysis for RW methods. Besides, we can show that RW methods have a higher probability to escape local minima than EW, resulting in better generalization performance. Empirically, we investigate lots of State-Of-The-Art (SOTA) task balancing approaches including four loss balancing methods and eight gradient balancing methods mostly under the multi-head architecture as those previous works did. On five Computer Vision (CV) datasets and two multilingual problems from the XTREME benchmark (Hu et al., 2020), we show that RW methods can consistently outperform EW and have competitive performance with existing SOTA methods.

In summary, the main contributions of this paper are three-fold.

- We propose the simple RW methods as novel baselines and litmus tests for MTL.

- We provide the convergence guarantee and effectiveness analysis for RW methods.

- Extensive experiments show that RW can outperform EW and achieve comparable performance with the SOTA methods.

## 2 Preliminary

**Notations.** Suppose there are $T$ tasks and task $t$ has its corresponding dataset $\mathcal{D}_t$. An MTL model usually contains two parts of parameters: task-sharing parameters $\theta$ and task-specific parameters $\{\psi_t\}_{t=1}^T$. For example, in CV, $\theta$ usually denotes parameters in the feature extractor shared by all the tasks and $\psi_t$ represents parameters in the task-specific output module for task $t$. Let $\ell_t(\mathcal{D}_t; \theta, \psi_t)$ denotes the average loss on $\mathcal{D}_t$ for task $t$. $\{\lambda_t^l\}_{t=1}^T$ are task-specific loss weights with a constraint that $\lambda_t^l \geq 0$ for all $t$'s. Similarly, $\{\lambda_t^g\}_{t=1}^T$ denote task-specific gradient weights.

**Conventional Baseline with Fixed Weights.** Since there are multiple losses in MTL, they usually are aggregated as a single one via loss weights as

$$\mathcal{L}(\theta, \{\psi_t\}_{t=1}^T) = \sum_{t=1}^T \lambda_t^l \ell_t(\mathcal{D}_t; \theta, \psi_t). \tag{1}$$

Apparently, the most simple method for loss weighting is to assign the same weight to all the tasks in the whole training process, i.e., without loss of generality, $\lambda_t^l = \frac{1}{T}$ for all $t$'s in each iteration. This approach is a common baseline in MTL and it is called EW in this paper.

**Loss Balancing Methods.** To achieve task balancing and improve the performance of MTL model, loss balancing methods aim to study how to generate appropriate loss weights $\{\lambda_t^l\}_{t=1}^T$ in Eq. (1) in each iteration and some representative methods include Uncertainty Weights (**UW**) (Kendall et al., 2018), Dynamic Weight Average (**DWA**) (Liu et al., 2019), **IMTL-L** (Liu et al., 2021b) and Multi-Objective Meta Learning

(**MOML**) (Ye et al., 2021). These four methods focus on using higher loss weights for more difficult tasks measured by the uncertainty, learning speed, relative loss value, and validation performance, respectively. A more detailed introduction of those loss balancing methods is put in Section 6. When minimizing Eq. (1), the learning rate of optimizing each task-specific parameter $\psi_t$ will be affected by the corresponding loss weight $\lambda_t^l$, which is the major difference between loss balancing and gradient balancing methods.

**Gradient Balancing Methods.** This type of method thinks that the task balancing problem is caused by conflicting task gradients and the inappropriate gradient to update task-sharing parameters, thus they solve it via generating appropriate gradient weights $\{\lambda_t^g\}_{t=1}^T$ to balance the task gradients and make a better update of $\theta$ in each iteration as

$$\theta = \theta - \eta \sum_{t=1}^T \lambda_t^g \nabla_\theta \ell_t(\mathcal{D}_t; \theta, \psi_t). \tag{2}$$

Noticeably, in such type methods, the gradient weights $\{\lambda_t^g\}_{t=1}^T$ only affect the task-sharing parameter $\theta$ but not task-specific parameters $\{\psi_t\}$, each of which is updated by the $t$-th task gradient $\nabla_{\psi_t} \ell_t(\mathcal{D}_t; \theta, \psi_t)$.

Some representative works include **MGDA-UB** (Sener & Koltun, 2018), Gradient Normalization (**GradNorm**) (Chen et al., 2018b), Projecting Conflicting Gradient (**PCGrad**) (Yu et al., 2020), Gradient sign Dropout (**GradDrop**) (Chen et al., 2020), Impartial Multi-Task Learning (**IMTL-G**) (Liu et al., 2021b), Gradient Vaccine (**GradVac**) (Wang et al., 2021), Conflict-Averse Gradient (**CAGrad**) (Liu et al., 2021a), and **RotoGrad** (Javaloy & Valera, 2022). Those eight methods focus on finding an aggregated gradient by linearly combining all the task gradients under different constraints such as equal gradient magnitude in GradNorm and equal gradient projection in IMTL-G to eliminate the gradient conflict. A more detailed introduction of those gradient balancing methods is put in Section 6.

Compared with the EW method, those two types of methods use a dynamic weighting process where loss/gradient weights vary over training iterations or epochs. Thus, it is natural to think how about training an MTL model with random weights. Inspired by this, we propose the RW methods by randomly sampling loss/gradient weights in each iteration as the random experiments for loss/gradient balancing methods, respectively. Besides, we think RW methods are more reasonable baselines than EW as the litmus tests for MTL methods.

## 3 The Random Weighting Methods

In this section, we introduce the RW methods, including the RLW and RGW methods.

We focus on the update of the task-sharing parameter $\theta$ as it is the key problem in MTL. In the following, we mainly introduce the RLW method as the RGW method acts similarly to the RLW method. For notation simplicity, we do not distinguish between $\lambda_t^l$ and $\lambda_t^g$ and denote them by $\lambda_t$. Besides, we denote $\boldsymbol{\ell}(\theta) = (\ell_1(\theta), \cdots, \ell_T(\theta))$, where the datasets $\{\mathcal{D}_t\}_{t=1}^T$ and the task-specific parameters $\{\psi_t\}_{t=1}^T$ are omitted for brevity.

Different from those loss balancing methods, RLW considers the loss weights $\boldsymbol{\lambda} = (\lambda_1, \cdots, \lambda_T) \in \mathbb{R}^T$ as random variables and samples them from a random distribution in each iteration. To guarantee loss weights in $\boldsymbol{\lambda}$ to be non-negative, we can first sample $\tilde{\boldsymbol{\lambda}} = (\tilde{\lambda}_1, \cdots, \tilde{\lambda}_T)$ from any distribution $p(\tilde{\boldsymbol{\lambda}})$ and then normalize $\tilde{\boldsymbol{\lambda}}$ into $\boldsymbol{\lambda}$ via a mapping $f$, where $f : \mathbb{R}^T \to \Delta^{T-1}$ is a normalization function such as the softmax function and $\Delta^{T-1}$ denotes a simplex in $\mathbb{R}^T$, i.e., $\boldsymbol{\lambda} \in \Delta^{T-1}$ means $\sum_{t=1}^T \lambda_t = 1$ and $\lambda_t \geq 0$ for all $t$. Note that $p(\boldsymbol{\lambda})$ is different from $p(\tilde{\boldsymbol{\lambda}})$ unless $f$ is an identity function. Finally, RLW updates the $\theta$ by computing the aggregated gradient $\nabla_\theta \boldsymbol{\lambda}^\top \boldsymbol{\ell}(\theta)$.

In this way, the RLW method uses dynamical loss weights in the training process, which is similar to existing loss balancing methods, but RLW uses random weights instead of carefully designed ones in the existing works. Therefore, RLW is a basic random experiment for those loss balancing methods to examine their effectiveness, which indicates RLW is a more reasonable baseline than the conventional EW.

---

**Algorithm 1** A training iteration in RW methods. The random sampling process is the only difference between RW methods and the existing works. The red line and blue line are the only difference between RLW and RGW methods.

---

1: **Input:** numbers of tasks $T$, learning rate $\eta$, dataset $\{\mathcal{D}_t\}_{t=1}^T$, weight distribution $p(\tilde{\boldsymbol{\lambda}})$, normalization function $f$
2: **Output:** task-sharing parameter $\theta'$, task-specific parameters $\{\psi'_t\}_{t=1}^T$
3: **for** $t = 1$ **to** $T$ **do**
4:     Compute loss $\ell_t(\mathcal{D}_t; \theta, \psi_t)$;
5:     Compute gradient $g_t = \nabla_\theta \ell_t(\mathcal{D}_t; \theta, \psi_t)$;
6: **end for**
7: Sample weights $\tilde{\boldsymbol{\lambda}}$ from $p(\tilde{\boldsymbol{\lambda}})$ and normalize it into $\boldsymbol{\lambda}$ via $f$;                ▷ Random Sampling
8: $\theta' = \theta - \eta \sum_{t=1}^T \lambda_t g_t$;
9: **for** $t = 1$ **to** $T$ **do**
10:     $\psi'_t = \psi_t - \eta \nabla_{\psi_t} \lambda_t \ell_t(\mathcal{D}_t; \theta, \psi_t)$ or $\psi'_t = \psi_t - \eta \nabla_{\psi_t} \ell_t(\mathcal{D}_t; \theta, \psi_t)$;
11: **end for**

---

Noticeably, the loss weights $\boldsymbol{\lambda}$ are random variables and vary over training iterations, thus it is apparent that the gradient $\nabla_\theta \boldsymbol{\lambda}^\top \boldsymbol{\ell}(\theta)$ of RLW is an unbiased estimation of the gradient $\mathbb{E}[\boldsymbol{\lambda}]^\top \nabla_\theta \boldsymbol{\ell}(\theta)$, where $\mathbb{E}[\boldsymbol{\lambda}]$ is the expectation of $\boldsymbol{\lambda}$ over the whole training process. This means that the RLW method is a stochastic variant of the loss balancing method with fixed weights $\mathbb{E}[\boldsymbol{\lambda}]$. In particular, if $\mathbb{E}[\boldsymbol{\lambda}]$ is proportional to $(\frac{1}{T}, \cdots, \frac{1}{T})$, RLW is a stochastic variant of the conventional EW baseline. In Section 4, we theoretically show that RLW has better generalization performance than EW because of the extra randomness from loss weight sampling, which indicates the RLW method is a more effective baseline than EW.

Similar to RLW, in each iteration, RGW first randomly samples gradient weights $\tilde{\boldsymbol{\lambda}}$ from $p(\tilde{\boldsymbol{\lambda}})$, then normalizes it to obtain $\boldsymbol{\lambda}$ via $f$, and finally updates the task-sharing parameter $\theta$ by computing the aggregated gradient $\nabla_\theta \boldsymbol{\lambda}^\top \boldsymbol{\ell}(\theta)$. Following previous works (Sener & Koltun, 2018; Chen et al., 2020; Liu et al., 2021b; Javaloy & Valera, 2022), we can also compute the gradient with respect to the final hidden feature representation $z$ output from the shared parameter instead of the task-sharing parameter $\theta$ to reduce the computational cost. Thus, RGW is a random experiment for gradient balancing methods.

In this paper, we use the standard normal distribution for $p(\tilde{\boldsymbol{\lambda}})$ and the softmax function for $f$ in both the RLW and RGW methods since it is easy to implement, has a more stable performance compared with other sampling distributions (as shown in experimental results in Section 5.4), and is as efficient as the EW strategy (as shown in experimental results in Section 5.5). Besides, $\mathbb{E}[\boldsymbol{\lambda}]$ is proportional to $(\frac{1}{T}, \cdots, \frac{1}{T})$ as proved in Appendix B, thus it is fair to compare with the EW strategy.

The training algorithms of both RW methods are summarized in Algorithm 1. The only difference between the RW methods and the existing works is the generation of loss/gradient weights (i.e., Line 7 in Algorithm 1). Apparently, the sampling operation in the RW methods is very easy to implement and only brings negligible additional computational costs when compared with the existing works. Note that random weights are involved in the update of task-specific parameters in the RLW method but not the RGW method (i.e., Line 10 in Algorithm 1).

## 4 Analysis

In this section, we analyze how the extra randomness from the loss/gradient weight sampling affects the convergence and effectiveness of the RW methods compared with the EW strategy.

We focus on the update of task-sharing parameter $\theta$ and take RLW as an example for analysis, which can easily be extended to the RGW method. For notation simplicity, we simply use $\ell_t(\theta)$ instead of $\ell_t(\mathcal{D}_t; \theta, \psi_t)$ to denote the loss function of task $t$, and define $\sigma_t^2 = \mathbb{E}_{\mathcal{D}_t}[\|\nabla \ell_t(\mathcal{D}_t; \theta)\|^2]$ and $\boldsymbol{\mu} = \mathbb{E}_{\boldsymbol{\lambda}}[\boldsymbol{\lambda}]$ in this section and Appendix A. For ease of analysis, we make the following assumptions.

**Assumption 1** (First Derivative Lipschitz Continuity). For each task $t$, the gradient $\nabla \ell_t(\theta)$ is $M_t$-Lipschitz continuous if for any two points $\theta_1$ and $\theta_2$, we have

$$\|\nabla \ell_t(\theta_1) - \nabla \ell_t(\theta_2)\| \leq M_t \|\theta_1 - \theta_2\|.$$

**Assumption 2** (Polyak-Lojasiewicz Condition). For each task $t$, the loss function $\ell_t(\theta)$ satisfies the Polyak-Lojasiewicz (PL) inequality with constant $c_t$ if for any $\theta$, we have

$$\frac{1}{2}\|\nabla \ell_t(\theta)\|^2 \geq c_t(\ell_t(\theta) - \ell_t^*),$$

where $\theta_t^*$ is a global minimum of $\ell_t(\theta)$ and $\ell_t^* = \ell_t(\theta_t^*)$ denotes the optimal function value of $\ell_t(\theta)$.

**Assumption 3** (One Point Strongly Convex). For each task $t$, the loss function $\ell_t(\theta)$ is $L_t$-one point strongly convex with respect to a minimum $\theta_*$ after convolved with noise $\xi$ if for any $\varphi$, we have

$$\langle \nabla \mathbb{E}_\xi \ell_t(\varphi - \eta\xi), \varphi - \theta_* \rangle \geq L_t \|\varphi - \theta_*\|^2.$$

Note that these three assumptions are widely used in the analysis of non-convex optimization problems (Vaswani et al., 2019; Safran et al., 2021), thus they could hold for deep MTL models where deep neural networks are used.

### 4.1 Convergence Analysis

In the following theorem, we analyze the convergence property of Algorithm 1 for the RLW method.

**Theorem 1** (Convergence). *Consider the RLW method with objective function $\mathcal{L}(\theta) = \boldsymbol{\mu}^\top \boldsymbol{\ell}(\theta)$. Suppose that Assumptions 1 and 2 holds. Define the global minimum of RLW as $\theta_* = \arg\min_\theta \mathcal{L}(\theta)$, and the solution in the $k$-th iteration in RLW method as $\theta_k$. If $\eta$, the step size or equivalently the learning rate, satisfies $\eta \leq 1/2c$, where $c = \min_{1 \leq t \leq T}\{c_t\}$, we have*

$$\mathbb{E}[\mathcal{L}(\theta_{k+1}) - \mathcal{L}(\theta_*)] \leq (1 - 2\eta c)\left[\mathcal{L}(\theta_k) - \mathcal{L}(\theta_*)\right] + \frac{M\eta^2\kappa}{2}, \tag{3}$$

*where $\kappa = \sum_{t=1}^{T} \sigma_t^2$ and $M = \max_{1 \leq t \leq T}\{M_t\}$. Then for any positive $\varepsilon$, $\mathbb{E}[\mathcal{L}(\theta_k) - \mathcal{L}(\theta_*)] \leq \varepsilon$ can be achieved after $k = \frac{M\kappa}{2\varepsilon c^2} \log\left(\frac{\varepsilon_0}{\varepsilon}\right)$ iterations with $\eta = \frac{2\varepsilon c}{M\kappa}$, where $\varepsilon_0 = \mathbb{E}[\mathcal{L}(\theta_0) - \mathcal{L}(\theta_*)]$.*

Theorem 1 shows that the RLW method with a fixed step size has a linear convergence up to a radius around the optimal solution, which is similar to the EW strategy according to the property of the standard Stochastic Gradient Descent (SGD) method (Moulines & Bach, 2011; Needell et al., 2016). Although the RLW method has a larger $\kappa$ than the EW strategy, i.e., $\kappa_{\text{EW}} = \sum_{t=1}^{T} \mu_t^2 \cdot \sum_{t=1}^{T} \sigma_t^2 \leq \kappa$, which may possibly require more iterations for the RLW method to reach the same accuracy as the EW strategy, experimental results in Section 5.5 show that empirically this does not cause much difference.

### 4.2 Effectiveness Analysis

Note that converging to flat local minima and escaping sharp local minima are important in neural network training because flat local minima may lead to better generalization (Keskar et al., 2017; Chaudhari et al., 2019). One popular way to escape sharp local minima is to inject noise into the gradient since sharp local minima has a smaller diameter than the flat one. The most representative work is SGD which can converge to a better solution than Gradient Descent (GD) techniques under various settings with the help of noisy gradients (Hardt et al., 2016; Kleinberg et al., 2018). Inspired by this, we analyze the effectiveness of the RLW method from the perspective of stochastic optimization as follows.

It is observed that for both EW and RLW methods, the stochastic gradient can be cast as a noise injected for the full gradient, i.e., the update step as

$$\theta_{k+1} = \theta_k - \eta(\nabla \boldsymbol{\mu}^\top \boldsymbol{\ell}(\theta_k) + \xi_k), \tag{4}$$

where $\xi_k$ is a noise with $\mathbb{E}[\xi_k] = 0$ and $\|\xi_k\|^2 \leq r$, and $r$ denotes the intensity of the noise. For EW method, the noise $\xi_k$ is caused by the randomness of data sampling. While for RLW method, the randomness is not only from data sampling but also from weights sampling, thus the RLW method can have a larger noise $\xi_k$ with a larger $r$ than the EW strategy (refer to Appendix A.3).

In the following Theorem 2, we show that under the MTL setting, both EW and RLW methods have high probabilities to converge to a local minimum $\theta_*$ with a radius $r$ due to the injected noise $\xi_k$ in the update step (i.e., Eq. (4)).

**Theorem 2.** *Suppose an intermediate sequence $\varphi_k = \theta_k - \eta\nabla\boldsymbol{\mu}^\top\boldsymbol{\ell}(\theta_k)$ and the update step of $\theta_k$ is as Eq. (4). Then under Assumptions 1 and 3, after $K = \frac{1}{\rho}\log\left(\frac{\rho\varepsilon_0}{\beta}\right)$ iterations with $\eta \leq \frac{L}{M^2}$, with probability at least $1 - \delta$, we have*

$$\|\varphi_K - \theta_*\|^2 \leq \frac{2\beta}{\rho\delta},$$

*where $\varepsilon_0 = \mathbb{E}[\|\varphi_0 - \theta_*\|^2]$, $L = \min_{1\leq t\leq T}\{L_t\}$, $M = \max_{1\leq t\leq T}\{M_t\}$, $\rho = 2\eta L - \eta^2 M^2$, and $\beta = \eta^2 r^2(1 + \eta M)^2$.*

Based on the definition of $\varphi_k$ and the update step of $\theta_k$ in Eq. (4), we have

$$\begin{aligned}
\mathbb{E}_{\xi_k}[\varphi_{k+1}] &= \varphi_k - \eta\nabla\mathbb{E}_{\xi_k}[\boldsymbol{\mu}^\top\boldsymbol{\ell}(\varphi_k - \eta\xi_k)] \\
&= \varphi_k - \eta\nabla\mathbb{E}_{\xi_k}[\boldsymbol{\mu}^\top\boldsymbol{\ell}(\theta_k - \eta(\nabla\boldsymbol{\mu}^\top\boldsymbol{\ell}(\theta_k) + \xi_k))] \\
&= \varphi_k - \eta\nabla\mathbb{E}_{\xi_k}[\boldsymbol{\mu}^\top\boldsymbol{\ell}(\theta_{k+1})].
\end{aligned}$$

Thus the sequence $\{\varphi_k\}$ can be regarded as an approximation of using GD to minimize the function $\mathbb{E}_{\xi_k}[\boldsymbol{\mu}^\top\boldsymbol{\ell}(\theta)]$, which indicates that if the sequence $\{\varphi_k\}$ converges to a local minimum $\theta_*$ according to Theorem 2, $\mathbb{E}_{\xi_k}[\boldsymbol{\mu}^\top\boldsymbol{\ell}(\theta)]$ can reach to the same minimum. Therefore, the parameters that lie in the neighborhoods of $\theta_*$ with a neighborhood size $r$ have similar loss function values. Since the RLW method has a larger $r$ than the EW method, thus RLW can converge to a flatter local minimum and achieve better generalization performance than EW.

## 5 Experiments

In this section, we empirically show the proposed RW methods can converge to a flatter minimum than the EW method in a toy example and then evaluate the proposed RW methods on five computer vision datasets (i.e., NYUv2, CityScapes, CelebA, Office-31, and Office-Home) and two multilingual problems from the XTREME benchmark (Hu et al., 2020). All the experiments are conducted on one single NVIDIA GeForce RTX 3090 GPU. The experimental results on the CityScapes, CelebA, Office-31, and Office-Home datasets are put in Appendix C.

**Compared Methods.** The baseline methods in comparison include several SOTA task balancing methods as introduced in Section 2, including four loss balancing methods (i.e., UW, DWA, IMTL-L, and MOML) and eight gradient balancing methods (i.e., MGDA-UB, GradNorm, PCGrad, GradDrop, IMTL-G, GradVac, CAGrad, and RotoGrad). For all the baseline methods, we directly use the optimal hyperparameters used in their original papers. The implementations of the RW methods and the baseline methods are based on the open-source LibMTL library (Lin & Zhang, 2022).

**Network Architecture.** The network architecture we used adopts the Hard-Parameter Sharing (**HPS**) pattern (Caruana, 1993), which shares the bottom layers of the network for all the tasks and uses separate top layers for each task. Other MTL architectures are studied in Section 5.7.

**Evaluation Metric.** For homogeneous MTL problems (e.g., the XTREME benchmark and Office-31 dataset) which contain tasks of the same type such as the classification task, we directly use the average performance among tasks as the performance metric. For heterogeneous MTL problems (e.g., the NYUv2 dataset) that contain tasks of different types and may have multiple evaluation metrics for each task, by

following (Maninis et al., 2019; Vandenhende et al., 2021), we use the average of the relative improvement over the EW method on each metric of each task as the performance measure, which is formulated as

$$\Delta_{\mathrm{p}} = 100\% \times \frac{1}{T} \sum_{t=1}^{T} \frac{1}{N_t} \sum_{n=1}^{N_t} \frac{(-1)^{p_{t,n}} (M_{t,n} - M_{t,n}^{\mathrm{EW}})}{M_{t,n}^{\mathrm{EW}}},$$

where $N_t$ denotes the number of metrics in task $t$, $M_{t,n}$ denotes the performance of a task balancing method for the $n$th metric in task $t$, $M_{t,n}^{\mathrm{EW}}$ is defined similarly for the EW method, and $p_{t,n}$ is set to 1 if a higher value indicates better performance for the $n$th metric in task $t$ and otherwise 0.

### 5.1 Results on Multi-Fashion-MNIST Datasets

Here, we take the RLW method as an example to empirically confirm whether the RW methods can converge to a flatter minimum than the EW method.

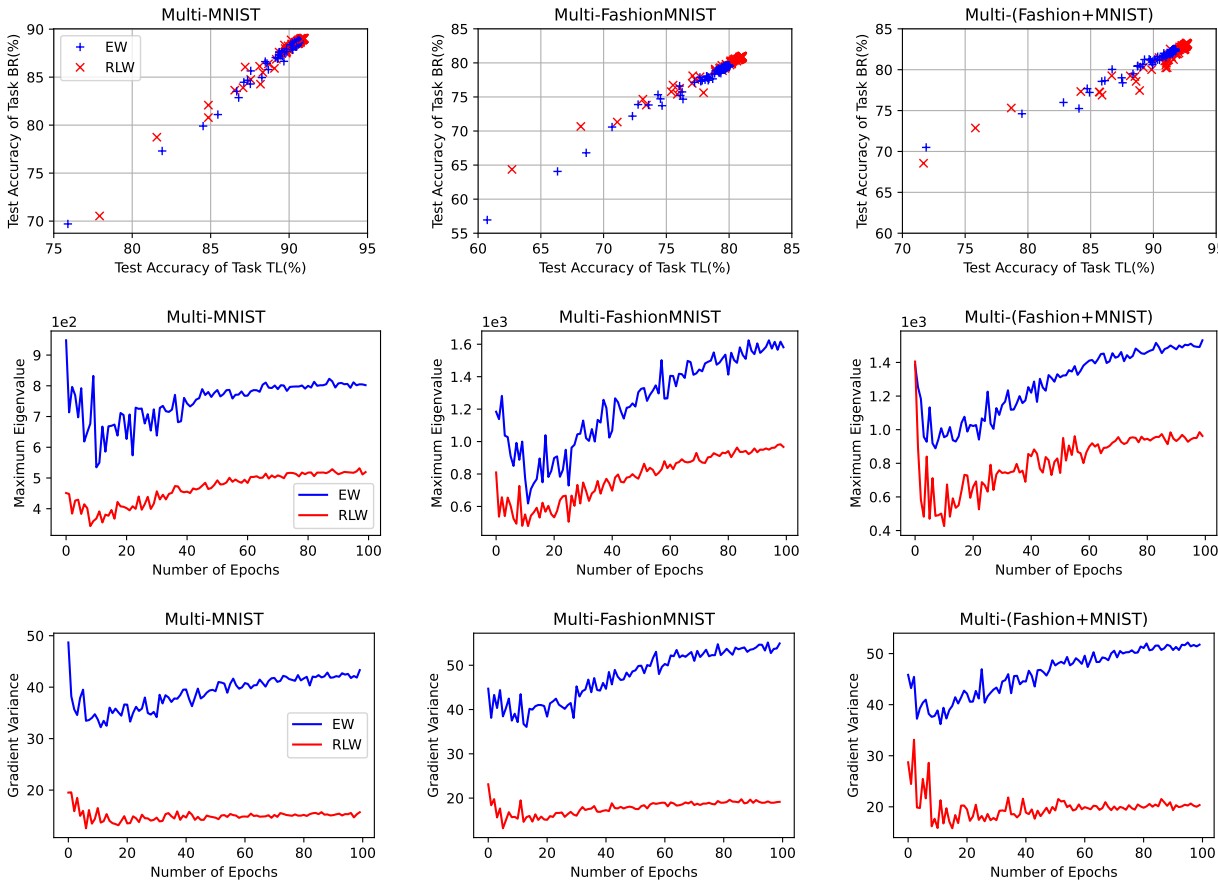

Figure 1: Results of the EW and RLW methods on the Multi-MNIST, the Multi-FashionMNIST, and the Multi-(Fashion+MNIST) datasets. Three figures in each column show the test accuracy of every task at each epoch, the variation of the maximum eigenvalue of Hessian within the training epoch, and the variation of the gradient variance within the training epoch on each dataset, respectively.

**Datasets.** We consider three image classifications datasets: the Multi-MNIST (Sabour et al., 2017), the Multi-FashionMNIST, and the Multi-(Fashion+MNIST) datasets (Lin et al., 2019). The Multi-MNIST dataset is generated by randomly sampling two images with different classes from the MNIST dataset (LeCun et al., 1998) and then overlaying one on top of another one to obtain a new image with size $36 \times 36$. This

dataset includes two tasks: classifying the digits on the top-left (**Task TL**) and bottom-right (**Task BR**), respectively. With the same approach, the Multi-FashionMNIST dataset is constructed with the overlap of the FashionMNIST images (Xiao et al., 2017), and the Multi-(Fashion+MNIST) dataset is constructed with the overlap of the FashionMNIST and MNIST images in (Lin et al., 2019). For both three datasets, we use 120K and 20K images for training and testing, respectively.

**Implementation Details.** These three datasets adopt the same experimental settings. Specifically, by following in (Lin et al., 2019), a LeNet-based network (LeCun et al., 1998) except the final fully connected layer is used as the shared encoder and a fully connected layer is used as the task-specific output layer. The SGD optimizer with the learning rate as $10^{-3}$ and the momentum as 0.9 is used for training, the batch size is set to 256, and the training epoch is set to 100. The cross-entropy loss is used for each task.

**Flatness Measures.** We compute two widely-used metrics to measure the flatness of a local minimum and the generalization for a deep neural network, i.e., the maximum eigenvalue of Hessian $\nabla^2 \mathcal{L}(\theta)$ computed on the whole training dataset (Foret et al., 2020) and the variance of gradient $\nabla \mathcal{L}(\theta)$ over all batches on the training dataset (Jiang et al., 2020). For both two metrics, a lower value means that the local minimum is flatter and the model has better generalization. Due to the high dimensionality of model parameter $\theta$, we calculate the approximated Hessian using the Lanczos algorithm (Ghorbani et al., 2019) based on the open-source implementation[1].

**Results.** The results are shown in Figure 1. The top row shows that the RLW method can significantly outperform the EW method on every task in these three datasets. The middle and last rows show that the RLW method has a smaller maximum eigenvalue and gradient variance than the EW method. Those results indicate that the RLW method can converge to a flatter local minimum and hence result in better generalization performance, which confirms the theoretical insight in Section 4.2.

## 5.2 Results on the NYUv2 Dataset

**Dataset.** The NYUv2 dataset (Silberman et al., 2012) is an indoor scene understanding dataset, which consists of video sequences recorded by the RGB and Depth cameras in the Microsoft Kinect. It contains 795 and 654 images for training and testing, respectively. This dataset includes three tasks: 13-class semantic segmentation, depth estimation, and surface normal prediction.

**Implementation Details.** For the NYUv2 dataset, the DeepLabV3+ architecture (Chen et al., 2018a) is used. Specifically, a ResNet-50 network pre-trained on the ImageNet dataset with dilated convolutions (Yu et al., 2017) is used as a shared encoder among tasks and the Atrous Spatial Pyramid Pooling (ASPP) (Chen et al., 2018a) module is used as the task-specific head for each task. Input images are resized to $288 \times 384$. The Adam optimizer (Kingma & Ba, 2015) with the learning rate as $10^{-4}$ and the weight decay as $10^{-5}$ is used for training and the batch size is set to 8. We use the cross-entropy loss, $L_1$ loss, and cosine loss as the loss function of the semantic segmentation, depth estimation, and surface normal prediction tasks, respectively.

**Results.** The results of different methods on the NYUv2 dataset are shown in Table 1. The top row shows the performance of the widely used EW strategy and we use it as a baseline to measure the relative improvement of different methods as shown in the definition of $\Delta_{\mathrm{p}}$. Rows 2-5 and 7-14 show the results of loss balancing and gradient balancing methods, respectively.

According to the results, we can see that both the RLW and RGW methods gain performance improvement over the EW strategy, which implies that training with extra randomness can have a better generalization ability. Besides, RLW has an improvement of 1.04% over the EW strategy and it is higher than all loss balancing methods. As for gradient balancing methods, half of those methods have negligible or even negative improvement over the EW strategy and RGW can outperform five of them. Compared with all

---

[1]`https://github.com/noahgolmant/pytorch-hessian-eigenthings`

Table 1: Performance on the **NYUv2** dataset with three tasks: 13-class semantic segmentation, depth estimation, and surface normal prediction. The best results for each task on each measure over loss/gradient balancing methods are marked with superscript $*/\dagger$. The best results for each task on each measure over all methods are highlighted in **bold**. $\uparrow$ ($\downarrow$) indicates that the higher (lower) the result, the better the performance.

| | Methods | Segmentation | | Depth | | Surface Normal | | | | | $\Delta_{\mathbf{p}}\uparrow$ |
| | | | | | | Angle Distance | | Within $t°$ | | | |
| | | mIoU↑ | Pix Acc↑ | Abs Err↓ | Rel Err↓ | Mean↓ | Median↓ | 11.25↑ | 22.5↑ | 30↑ | |
|---|---|---|---|---|---|---|---|---|---|---|---|
| | EW | 53.77 | 75.45 | 0.3845 | 0.1605 | 23.5737 | 17.0438 | 35.04 | 60.93 | 72.07 | +0.00% |
| Loss Bal. | UW | 54.14 | 75.92 | 0.3833 | 0.1597 | 23.2989 | 16.8691 | 35.33 | 61.37 | 72.48 | +0.64% |
| | DWA | 53.81 | 75.56 | 0.3792* | 0.1565* | 23.6111 | 17.0609 | 34.89 | 60.89 | 71.97 | +0.63% |
| | IMTL-L | 53.50 | 75.18 | 0.3824 | 0.1596 | 23.3805 | 16.8088 | 35.44 | 61.43 | 72.43 | +0.35% |
| | MOML | **54.98*** | **75.98*** | 0.3877 | 0.1618 | 23.2401* | 16.7388 | 35.90* | 61.81* | 72.76* | +0.76% |
| | **RLW (ours)** | 54.11 | 75.77 | 0.3809 | 0.1575 | 23.3777 | 16.7385* | 35.71 | 61.52 | 72.45 | +1.04%* |
| Gradient Bal. | MGDA-UB | 50.42 | 73.46 | 0.3834 | **0.1555**$^\dagger$ | **22.7827**$^\dagger$ | **16.1432**$^\dagger$ | **36.90**$^\dagger$ | 62.88 | 73.61 | +0.38% |
| | GradNorm | 53.58 | 75.06 | 0.3931 | 0.1663 | 23.4360 | 16.9844 | 35.11 | 61.11 | 72.24 | -0.99% |
| | PCGrad | 53.70 | 75.41 | 0.3903 | 0.1607 | 23.4281 | 16.9699 | 35.16 | 61.19 | 72.28 | -0.16% |
| | GradDrop | 53.58 | 75.56 | 0.3855 | 0.1592 | 23.5518 | 17.0137 | 35.08 | 60.97 | 72.02 | +0.08% |
| | IMTL-G | 53.54 | 75.45 | 0.3880 | 0.1589 | 23.0530 | 16.4328 | 36.21 | 62.31 | 73.15 | +0.80% |
| | GradVac | 54.89$^\dagger$ | **75.98**$^\dagger$ | 0.3828 | 0.1635 | 23.6865 | 17.1301 | 34.82 | 60.71 | 71.81 | +0.07% |
| | CAGrad | 53.12 | 75.19 | 0.3871 | 0.1599 | 22.5257 | 15.8821 | 37.42 | **63.50**$^\dagger$ | **74.17**$^\dagger$ | **+1.36%**$^\dagger$ |
| | RotoGrad | 53.90 | 75.46 | 0.3812 | 0.1596 | 23.0197 | 16.3714 | 36.37 | 62.28 | 73.05 | +1.19% |
| | **RGW (ours)** | 53.85 | 75.87 | **0.3772**$^\dagger$ | 0.1562 | 23.6725 | 17.2439 | 34.62 | 60.49 | 71.75 | +0.62% |

baselines, RLW is even higher than all of them except the CAGrad and RotoGrad methods, which indicates the random weights can easily beat the carefully designed ones.

According to the above analysis, there are two important conclusions. Firstly, the conventional EW strategy is a weaker baseline than RLW and RGW for MTL. Secondly, RW methods are competitive with SOTA methods and even perform better than some of them.

### 5.3 Results on the XTREME benchmark

**Dataset.** The XTREME benchmark (Hu et al., 2020) is a large-scale multilingual multi-task benchmark for cross-lingual generalization evaluation, which covers fifty languages and contains nine tasks. We conduct experiments on two tasks containing Paraphrase Identification (PI) and Part-Of-Speech (POS) tagging in this benchmark. The datasets used in the PI and POS tasks are the PAWS-X dataset (Yang et al., 2019) and Universal Dependency v2.5 treebanks (Nivre et al., 2020), respectively. On each task, we construct a multilingual problem by choosing the four languages with largest numbers of data, i.e., English (`en`), Mandarin (`zh`), German (`de`) and Spanish (`es`), for the PI task and English, Mandarin, Telugu (`te`) and Vietnamese (`vi`) for the POS task. The statistics for each language are summarized in Table 2. Different from the NYUv2 dataset where different tasks share the same input data, in those multilingual problems, each language/task has its own input data.

Table 2: The numbers of training, validation, and test data for each language in PI and POS problems from the XTREME benchmark.

| | PI | POS |
|---|---|---|
| `en` | 49.4K+2.0K+2.0K | 6.9K+1.8K+3.2K |
| `zh` | 49.4K+2.0K+2.0K | 4.0K+0.5K+2.9K |
| `de` | 49.4K+2.0K+2.0K | - |
| `es` | 49.4K+2.0K+2.0K | - |
| `te` | - | 1.0K+0.1K+0.1K |
| `vi` | - | 1.4K+0.8K+0.8K |

**Implementation Details.** For each multilingual problem in the XTREME benchmark, a pre-trained multilingual BERT (mBERT) model (Devlin et al., 2019) implemented via the open-source transformers library (Wolf et al., 2020) is used as the shared encoder among languages and a fully connected layer is used as the language-specific output layer for each language. The Adam optimizer with the learning rate as $2 \times 10^{-5}$ and the weight decay as $10^{-8}$ is used for training and the batch size is set to 32. The cross-entropy loss is used for the two multilingual problems.

**Results.** According to experimental results shown in Table 3, we can find some empirical observations, which are similar to those on the NYUv2 dataset. Firstly, both the RLW and RGW strategies outperform the EW method. Secondly, compared with the existing works, RLW and RGW can achieve comparable performance with existing loss/gradient balancing methods, respectively. Even, RLW or RGW methods could outperform all baseline methods. For example, RLW achieves the best performance (i.e., 90.25% average accuracy) on the PI problem and RGW achieves the best average F1 score of 91.16% on the POS problem. It is interesting to find that the performance of RLW and RGW are inconsistent in different datasets. There is because the random loss weights in RLW will affect the update of task-specific parameters while not in RGW, which has a different influence on the performance of different datasets.

Table 3: Performance on two multilingual problems, i.e., PI and POS from the **XTREME benchmark**. The best results for each language over loss/gradient balancing methods are marked with superscript $*/\dagger$. The best results for each language over all methods are highlighted in **bold**.

| | Methods | PI (Accuracy) | | | | | POS (F1 Score) | | | | |
|---|---|---|---|---|---|---|---|---|---|---|---|
| | | en | zh | de | es | **Avg** | en | zh | te | vi | **Avg** |
| | EW | 94.29 | 84.99 | 89.79 | 90.94 | 90.00 | 95.06 | 89.01 | 91.41 | 86.65 | 90.53 |
| Loss Bal. | UW | 93.74 | 85.44* | **90.24*** | 91.29 | 90.18 | 94.89 | 88.77 | 90.96 | 87.12 | 90.44 |
| | DWA | **94.69*** | 84.99 | 89.49 | **91.44*** | 90.15 | 95.02 | 89.03 | 91.87 | **87.27*** | 90.80 |
| | IMTL-L | 93.94 | 84.54 | 89.39 | **91.44*** | 89.82 | **95.57*** | 89.93* | 91.77 | 86.11 | 90.84 |
| | MOML | 93.89 | 83.74 | 89.94 | 90.99 | 89.64 | 95.15 | 89.11 | 92.41 | 87.24 | 90.98* |
| | **RLW (ours)** | 94.29 | 85.39 | 89.94 | 91.39 | **90.25*** | 95.01 | 88.87 | **92.86*** | 86.85 | 90.90 |
| Gradient Bal. | MGDA-UB | 94.09 | 84.14 | 89.14 | 90.59 | 89.49 | 94.89 | 88.43 | 91.01 | 86.04 | 90.01 |
| | GradNorm | 94.19 | 83.59 | 88.89 | 91.24 | 89.47 | 94.88 | 88.80 | 91.78 | 86.96 | 90.61 |
| | PCGrad | 94.19 | 85.49$^\dagger$ | 89.09 | 91.24 | 90.00 | 94.85 | 88.42 | 90.72 | 86.71 | 90.18 |
| | GradDrop | 94.29 | 84.44 | 89.69 | 90.94 | 89.84 | 95.08 | 89.06 | 90.65 | 87.17 | 90.49 |
| | IMTL-G | 94.69$^\dagger$ | 84.54 | 89.39 | 90.69 | 89.82 | 94.93 | 88.70 | 91.66 | 87.00 | 90.57 |
| | GradVac | 94.29 | 84.94 | 89.19 | 90.89 | 89.83 | 94.87 | 88.41 | 90.62 | 86.47 | 90.09 |
| | CAGrad | 94.34 | 84.59 | 90.09$^\dagger$ | 90.64 | 89.91 | 94.83 | 88.65 | 91.71 | 86.76 | 90.48 |
| | RotoGrad | 93.99 | 83.89 | 89.29 | 90.94 | 89.52 | 95.44 | 89.79 | 91.42 | 86.33 | 90.74 |
| | **RGW (ours)** | 94.55 | 84.99 | 89.29 | 91.40$^\dagger$ | 90.06$^\dagger$ | 95.52$^\dagger$ | **90.13$^\dagger$** | 91.82$^\dagger$ | 87.18$^\dagger$ | **91.16$^\dagger$** |

### 5.4 Robustness on Sampling Distribution

In this section, we evaluate the robustness of the proposed RW methods on the sampling distribution. Taking RLW as an example, we show its robustness by evaluating with five different sampling distributions (i.e., $p(\tilde{\boldsymbol{\lambda}})$) for loss weights. The five distributions are uniform distribution between 0 and 1 (denoted by **Uniform**), standard normal distribution (denoted by **Normal**), Dirichlet distribution with $\alpha = 1$ (denoted by **Dirichlet**), Bernoulli distribution with probability 1/2 (denoted by **Bernoulli**), Bernoulli distribution with probability 1/2 and a constraint $\sum_{t=1}^{T} \tilde{\lambda}_t = 1$ (denoted by **c-Bernoulli**). We set $f$ as a function of $f(\tilde{\boldsymbol{\lambda}}) = \tilde{\boldsymbol{\lambda}}/(\sum_{t=1}^{T} \tilde{\lambda}_t)$ for the Bernoulli distribution and the c-Bernoulli distribution, a softmax function for the Normal distribution and Uniform distribution, and an identity function for the Dirichlet distribution. We can prove that all the $\mathbb{E}[\boldsymbol{\lambda}]$'s under these five distributions equal $(\frac{1}{T}, \cdots, \frac{1}{T})$ (refer to Appendix B), thus it is fair to compare among them.

Figure 2 shows the results of the RLW method with five sampling distributions on the NYUv2 dataset in terms of $\Delta_{\mathrm{p}}$, where the experiment on each sampling distribution is repeated for 8 times. The results show

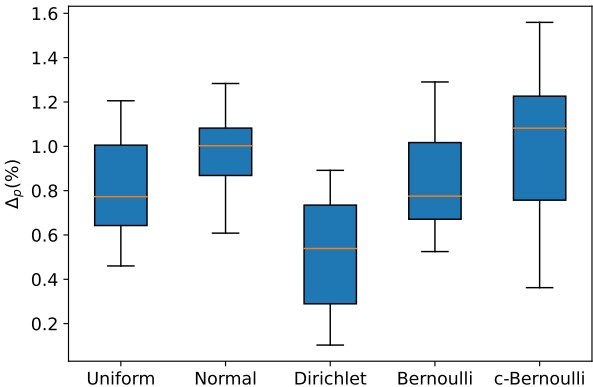

Figure 2: Results of the RLW method with different sampling distributions in terms of $\Delta_{\mathrm{p}}$.

that the RLW method with different distributions can always outperform the EW method, which shows the robustness of the RLW method with respect to the sampling distribution. In addition, compared with the uniform, Dirichlet, and Bernoulli distributions, RLW with the standard normal distribution achieves better and more stable performance. Although RLW with the c-Bernoulli distribution performs slightly better than the standard normal distribution, it is more unstable and may need a longer training time as shown in Section 5.5. Thus, in this paper, we use the standard normal distribution to sample loss weights.

## 5.5 Convergence Speed

Here we take RLW as an example to show the efficiency of RW methods. Figure 3 plots the performance curve on both NYUv2 and CelebA validation datasets to empirically compare the convergence speed of the EW and RLW methods.

On the NYUv2 dataset with 3 tasks, the performance curves of the RLW method with two sampling distributions are similar to that of the EW method, which indicates that the RLW method has a similar convergence property to the EW method on this dataset. As the number of tasks increases, i.e., on the CelebA dataset with 40 tasks, we find that the RLW method with the standard normal distribution still converges as fast as the EW method, while the RLW method with the c-Bernoulli distribution converges slower. One reason for this phenomenon is that only one task is used to update model parameters in each training iteration when using the c-Bernoulli distribution. Thus, in this paper, we use the standard normal distribution, which is as efficient as the EW method.

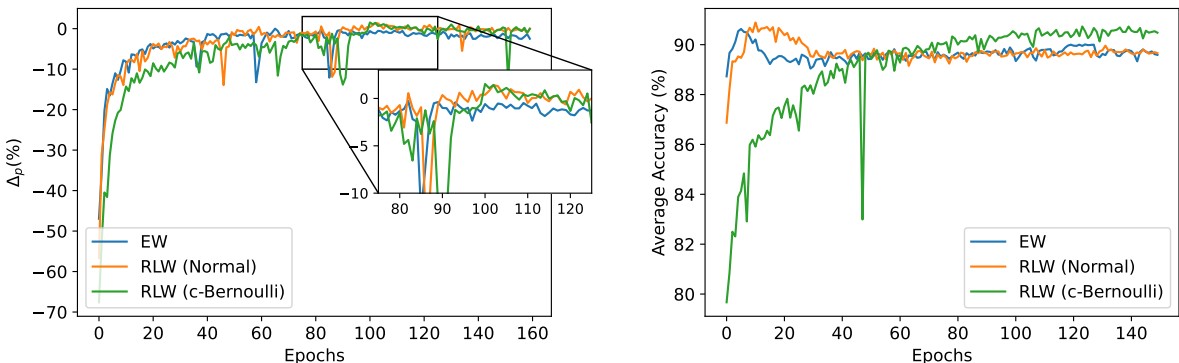

Figure 3: Comparison on the convergence speed of the EW and RLW methods on the 3-task NYUv2 validation dataset (**Left**) and the 40-task CelebA validation dataset (**Right**). We lean to use the standard normal distribution in the RLW method since it is as efficient as the EW method.

## 5.6 Combination of Loss and Gradient Balancing Methods

The loss balancing methods are complementary to the gradient balancing methods. Following (Liu et al., 2021b), we train an MTL model with different combinations of loss balancing and gradient balancing methods on the NYUv2 dataset to further improve the performance. We use the vanilla EW as the baseline to measure the relative improvement of the other different combinations as shown in the definition of $\Delta_p$.

According to the results shown in Table 4, we can see that combined with the UW, DWA, and IMTL-L methods, some gradient balancing methods performs better but others become worse. For example, $\Delta_p$ of the GradDrop method drops from $0.08\%$ to $-0.42\%$ when combined with DWA. Noticeably, by combining with the proposed RLW method, all the gradient balancing methods can achieve better performance. Besides, on each gradient balancing method, the improvement induced by the RLW method is significantly larger than the other three loss balancing methods as well as the EW method. Moreover, RGW can also improve the performance of loss balancing methods except for DWA. Thus, this experiment further demonstrates the effectiveness of the proposed RW methods.

Table 4: Results of different combinations of loss balancing and gradient balancing methods on the NYUv2 dataset in terms of $\Delta_p$. The best results in each row are highlighted in **bold**.

|  | EW | UW | DWA | IMTL-L | **RLW** |
|---|---|---|---|---|---|
| Vanilla | +0.00% | +0.64% | +0.63% | +0.35% | **+1.04%** |
| MGDA-UB | +0.38% | +0.15% | +0.47% | -0.59% | **+2.01%** |
| GradNorm | -0.99% | +0.87% | -0.95% | +0.54% | **+0.89%** |
| PCGrad | -0.16% | +0.72% | +0.19% | +0.38% | **+0.97%** |
| GradDrop | +0.08% | +0.25% | -0.42% | +0.36% | **+0.93%** |
| IMTL-G | +0.80% | +0.45% | +1.20% | +0.18% | **+1.50%** |
| GradVac | +0.07% | -0.03% | +0.89% | +0.69% | **+0.97%** |
| CAGrad | +1.36% | +1.07% | +1.41% | +2.18% | **+2.20%** |
| RotoGrad | +1.19% | +1.03% | +0.75% | +1.40% | **+1.45%** |
| **RGW** | +0.62% | +0.82% | +0.41% | +0.78% | **+1.46%** |

## 5.7 Effects of Different Architectures

The proposed RW methods can be seamlessly incorporated into all the MTL architectures. To see this, we take RLW as an example and combine it with three different MTL architectures, i.e., **cross-stitch** network (Misra et al., 2016), Multi-Task Attention Network (**MTAN**) (Liu et al., 2019), and **NDDR-CNN** (Gao et al., 2019). We use the combination of EW and HPS as the baseline to measure the relative improvement of the other different combinations as shown in the definition of $\Delta_p$.

Table 5: Results of different combinations of task balancing methods and MTL architectures on the NYUv2 dataset in terms of $\Delta_p$. The best results for each architecture are highlighted in **bold**.

|  | HPS | Cross-stitch | MTAN | NDDR-CNN |
|---|---|---|---|---|
| EW | +0.00% | +1.43% | +2.56% | +1.90% |
| CAGrad | +1.36% | +2.42% | +2.26% | +2.83% |
| **RLW** | +1.04% | +2.23% | +2.66% | +2.91% |
| **RLW**+CAGrad | **+2.20%** | **+2.76%** | **+2.92%** | **+3.53%** |

According to the results on the NYUv2 dataset as shown in Table 5, we can see that the proposed RLW strategy outperforms the EW method under all the three architectures. When using the MTAN and NNDR-CNN architectures, RLW achieves better performance than the CAGrad method that performs best in the HPS architecture, which shows the potential of the proposed RLW method when choosing suitable MTL architectures. Moreover, combined with the RLW method, CAGrad can be further improved under the four

architectures. For example, the combinations of RLW and CAGrad can achieve the best $\Delta_p$ of 3.53% under the NDDR-CNN architecture.

## 6 Related Work

**Loss Balancing Methods.** This type of method aims to balance different tasks by carefully designing the dynamic loss weights via measuring the learning speed, uncertainty, relative loss value, and validation performance. For example, **DWA** (Liu et al., 2019) sets the loss weight of each task to be the ratio of two adjacent losses. **UW** (Kendall et al., 2018) applies homoscedastic uncertainties as loss weights for each task and those weights are dynamically updated by backpropagation. **IMTL-L** (Liu et al., 2021b) learns the loss weights to make the scaled loss value is similar for each task. **MOML** (Ye et al., 2021) learns loss weights in a meta-learning-based hyperparameter optimization manner, where the loss weights are cast as the function of model parameters and updated by minimizing the loss on the validation dataset.

**Gradient Balancing Methods.** This type of method aims to find an aggregated gradient to balance different tasks. For example, **MGDA-UB** (Sener & Koltun, 2018) casts MTL as a multi-objective optimization problem and directly solve the optimal weights in every iteration by applying Multiple Gradient Descent Algorithm (MGDA) (Désidéri, 2012) which aims to find a common descending direction among all the gradients via solving a quadratic programming problem. **GradNorm** (Chen et al., 2018b) aims to explicitly learn loss weights by constraining the gradient magnitude of each task to be similar. **GradDrop** (Chen et al., 2020) thinks that the conflict can be reflected by the inconsistency in signs between the gradient values of different tasks and it eliminates such conflict by masking out the gradient values with opposite signs. **PCGrad** (Yu et al., 2020) projects each task's gradient onto the normal plane of the other task's gradient if they are conflict reflected in terms of negative cosine similarities between task gradients. Based on PCGrad, **GradVac** (Wang et al., 2021) projects task gradients in a more adaptive way where the cosine similarities between task gradients is not always positive. **IMTL-G** (Liu et al., 2021b) aims to find an aggregated gradient that has projections with an equal length on each task's gradient. **CAGrad** (Liu et al., 2021a) aims to find a gradient that can decrease all the task losses by solving an optimization problem at every iteration. **RotoGrad** (Javaloy & Valera, 2022) eliminates conflict by jointly homogenizing gradient magnitudes and directions. Specifically, it uses a learnable rotation matrix to change the gradient direction for each task and then computes weights that constrain all tasks to have the same gradient magnitudes.

**Task Similarity-based Methods.** This type of MTL method aims to improve the performance of MTL by modeling and utilizing the task similarities between different tasks. For example, Shui et al. (2019) explicitly learn a similarity matrix to model the similarities between different tasks in an adversarial manner and this adversarial loss term has a regularization effect to improve the performance. Similarly, Bai & Zhao (2022) propose a regularizer to learn the task similarities and improve the performance. Different from these two methods, Fifty et al. (2021) aim to automatically determine task grouping using the learned task similarities.

## 7 Conclusions

In this paper, we propose the RW methods, an important yet ignored baseline for MTL, by training an MTL model with random loss/gradient weights. We analyze the convergence and effectiveness properties of the proposed RW method. Moreover, we provide a consistent and comparative comparison to show that the RW methods can achieve comparable performance with state-of-the-art methods that use carefully designed weights. Though RW methods can not always achieve the best performance empirically, those experimental results still indicate that random experiments could be used to examine the effectiveness of newly proposed MTL methods and RW methods cou attract wide attention as the litmus tests. In our future work, we will apply the RW methods to more MTL applications.

## Acknowledgements

This work is supported by NSFC key grant 62136005, NSFC general grant 62076118, and Shenzhen fundamental research program JCYJ20210324105000003.

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

# A    Proof of Section 4

**Corollary 1.** *Consider the objective function $\mathcal{L}(\theta)$ defined in Theorem 1. Suppose that Assumption 1 is satisfied, then the function $\nabla\mathcal{L}(\theta)$ is $M$-Lipschitz continuous, where $M = \max_{1\le t\le T}\{M_t\}$.*

*Proof.* Note that $0 \le \mu_t \le 1$ and $\sum_{t=1}^{T}\mu_t = 1$, we have $\sum_{t=1}^{T}\mu_t M_t \le M$, where $M = \max_{1\le t\le T}\{M_t\}$. Since $\nabla\ell_t(\theta)$ is $M_t$-Lipschitz continuous, for any two points $\theta_1$ and $\theta_2$, we have

$$
\begin{aligned}
\|\nabla\mathcal{L}(\theta_1) - \nabla\mathcal{L}(\theta_2)\| &= \left\|\nabla\sum_{t=1}^{T}\mu_t\ell_t(\theta_1) - \nabla\sum_{t=1}^{T}\mu_t\ell_t(\theta_2)\right\| \\
&= \sum_{t=1}^{T}\mu_t\|\nabla\ell_t(\theta_1) - \nabla\ell_t(\theta_2)\| \\
&\le \left(\sum_{t=1}^{T}\mu_t M_t\right)\|\theta_1 - \theta_2\| \\
&\le M\|\theta_1 - \theta_2\|.
\end{aligned}
$$

Thus $\nabla\mathcal{L}(\theta)$ is $M$-Lipschitz continuous.    □

**Corollary 2.** *Suppose that Assumption 2 is satisfied, then the function $\mathcal{L}(\theta)$ defined in Theorem 1 satisfies the PL inequality with constant $c$, where $c = \min_{1\le t\le T}\{c_t\}$.*

*Proof.* Suppose $c = \min_{1\le t\le T}\{c_t\}$ and $\theta_*$ is a global minimum of $\mathcal{L}(\theta)$. Since $\ell_t(\theta)$ satisfies the PL inequality with constant $c_t$, for any $\theta$, we have

$$
\begin{aligned}
\frac{1}{2}\|\nabla\mathcal{L}(\theta)\|^2 &= \frac{1}{2}\left\|\nabla\sum_{t=1}^{T}\mu_t\ell_t(\theta)\right\|^2 \\
&\ge \frac{1}{2}\sum_{t=1}^{T}\mu_t\|\nabla\ell_t(\theta)\|^2 \\
&\ge c\sum_{t=1}^{T}\mu_t(\ell_t(\theta) - \ell_t(\theta_t^*)),
\end{aligned}
$$

where the last inequality is due to the PL conditions in Assumption 2 and the definition of $c$.

Since the optimal value of the weighted objective $\mathcal{L}(\theta)$ is no less than the weighting of the respective optimal values of all objectives $\ell_t(\theta_t^*)$. We have $\mathcal{L}(\theta_*) \ge \sum_{t=1}^{T}\mu_t\ell_t(\theta_t^*)$. So we obtain

$$
\frac{1}{2}\|\nabla\mathcal{L}(\theta)\|^2 \ge c(\mathcal{L}(\theta) - \mathcal{L}(\theta_*)).
$$

Thus $\mathcal{L}(\theta)$ satisfies the PL inequality with the constant $c$.    □

## A.1    Proof of Theorem 1

Based on Corollary 1 and the update of stochastic gradient descent, i.e., $\theta_{k+1} = \theta_k - \eta\nabla\boldsymbol{\lambda}^{\top}\boldsymbol{\ell}(\tilde{\boldsymbol{\mathcal{D}}};\theta_k)$, we have

$$
\begin{aligned}
\mathcal{L}(\theta_{k+1}) - \mathcal{L}(\theta_k) &\le \langle\nabla\mathcal{L}(\theta_k), \theta_{k+1} - \theta_k\rangle + \frac{M}{2}\|\theta_{k+1} - \theta_k\|^2 \\
&= -\eta\left\langle\nabla\mathcal{L}(\theta_k), \nabla\boldsymbol{\lambda}^{\top}\boldsymbol{\ell}(\tilde{\boldsymbol{\mathcal{D}}};\theta_k)\right\rangle + \frac{M\eta^2}{2}\|\nabla\boldsymbol{\lambda}^{\top}\boldsymbol{\ell}(\tilde{\boldsymbol{\mathcal{D}}};\theta_k)\|^2.
\end{aligned}
$$

Note that $\mathbb{E}_{\boldsymbol{\lambda}}\left[\mathbb{E}_{\tilde{\boldsymbol{\mathcal{D}}}}[\nabla\boldsymbol{\lambda}^{\top}\boldsymbol{\ell}(\tilde{\boldsymbol{\mathcal{D}}};\theta_k)]\right] = \nabla\boldsymbol{\mu}^{\top}\boldsymbol{\ell}(\boldsymbol{\mathcal{D}};\theta_k)$ and

$$\mathbb{E}_{\boldsymbol{\lambda}}\left[\mathbb{E}_{\tilde{\boldsymbol{\mathcal{D}}}}[\|\nabla\boldsymbol{\lambda}^{\top}\boldsymbol{\ell}(\tilde{\boldsymbol{\mathcal{D}}};\theta_k)\|^2]\right] \le \mathbb{E}_{\boldsymbol{\lambda}}\left[\mathbb{E}_{\tilde{\boldsymbol{\mathcal{D}}}}[\|\boldsymbol{\lambda}^{\top}\|^2\|\nabla\boldsymbol{\ell}(\tilde{\boldsymbol{\mathcal{D}}};\theta_k)\|^2]\right]$$

$$\le \mathbb{E}_{\boldsymbol{\lambda}}\left[\sum_{t=1}^{T}\lambda_t^2\right]\cdot\sum_{t=1}^{T}\sigma_t^2$$

$$\le \sum_{t=1}^{T}\sigma_t^2,$$

where the first inequality is due to the Cauchy-Schwarz inequality and the third inequality is due to $0 \le \lambda_t \le 1$ and $\sum_t \lambda_t = 1$. Then, by defining $\kappa = \sum_{t=1}^{T}\sigma_t^2$, we obtain

$$\mathbb{E}_{\boldsymbol{\lambda}}\left[\mathbb{E}_{\tilde{\boldsymbol{\mathcal{D}}}}[\mathcal{L}(\theta_{k+1}) - \mathcal{L}(\theta_k)]\right] = -\eta\left\langle\nabla\mathcal{L}(\theta_k), \mathbb{E}_{\boldsymbol{\lambda}}\left[\mathbb{E}_{\tilde{\boldsymbol{\mathcal{D}}}}[\nabla\boldsymbol{\lambda}^{\top}\boldsymbol{\ell}(\tilde{\boldsymbol{\mathcal{D}}};\theta_k)]\right]\right\rangle + \frac{M\eta^2}{2}\mathbb{E}_{\boldsymbol{\lambda}}\left[\mathbb{E}_{\tilde{\boldsymbol{\mathcal{D}}}}[\|\nabla\boldsymbol{\lambda}^{\top}\boldsymbol{\ell}(\tilde{\boldsymbol{\mathcal{D}}};\theta_k)\|^2]\right]$$

$$\le -\eta\left\langle\nabla\mathcal{L}(\theta_k), \nabla\boldsymbol{\mu}^{\top}\boldsymbol{\ell}(\boldsymbol{\mathcal{D}};\theta_k)\right\rangle + \frac{M\eta^2\kappa}{2}$$

$$= -\eta\|\nabla\boldsymbol{\mu}^{\top}\boldsymbol{\ell}(\theta_k)\|^2 + \frac{M\eta^2\kappa}{2}$$

$$\le -2\eta c(\mathcal{L}(\theta_k) - \mathcal{L}(\theta_*)) + \frac{M\eta^2\kappa}{2},$$

where the last inequality is due to Corollary 2. Then, we have

$$\mathbb{E}[\mathcal{L}(\theta_{k+1}) - \mathcal{L}(\theta_*)] \le (1 - 2\eta c)(\mathcal{L}(\theta_k) - \mathcal{L}(\theta_*)) + \frac{M\eta^2\kappa}{2}. \tag{5}$$

If $1 - 2\eta c > 0$, we recursively apply the inequality (5) over the first $k$ iterations and we can obtain

$$\mathbb{E}[\mathcal{L}(\theta_{k+1}) - \mathcal{L}(\theta_*)] \le (1 - 2\eta c)^k(\mathcal{L}(\theta_0) - \mathcal{L}(\theta_*)) + \frac{M\eta^2\kappa}{2}\sum_{j=0}^{k-1}(1 - 2\eta c)^j$$

$$\le (1 - 2\eta c)^k(\mathcal{L}(\theta_0) - \mathcal{L}(\theta_*)) + \frac{M\eta\kappa}{4c}.$$

Thus the inequality (3) holds if $\eta \le \frac{1}{2c}$.

According to inequality (5), the minimal value of a quadratic function $g_\varepsilon(\eta) = (1 - 2\eta c)\varepsilon + \frac{M\eta^2\kappa}{2}$ is achieved at $\eta_* = \frac{2\varepsilon c}{M\kappa}$. By setting $\mathcal{L}(\theta_k) - \mathcal{L}(\theta_*) = \varepsilon_0$, we have

$$\mathbb{E}[\mathcal{L}(\theta_{k+1}) - \mathcal{L}(\theta_*)] \le g_{\mathcal{L}(\theta_k) - \mathcal{L}(\theta_*)}(\eta_*)$$

$$= \left(1 - \frac{2c^2[\mathcal{L}(\theta_k) - \mathcal{L}(\theta_*)]}{M\kappa}\right)[\mathcal{L}(\theta_k) - \mathcal{L}(\theta_*)]$$

$$\le \left(1 - \frac{2\varepsilon c^2}{M\kappa}\right)[\mathcal{L}(\theta_k) - \mathcal{L}(\theta_*)]$$

$$\le \left(1 - \frac{2\varepsilon c^2}{M\kappa}\right)^k\varepsilon_0.$$

Then if $\mathbb{E}[\mathcal{L}(\theta_{k+1}) - \mathcal{L}(\theta_*)] \ge \varepsilon$, we have $\varepsilon \le (1 - \frac{2\varepsilon c^2}{M\kappa})^k\varepsilon_0$. Therefore, $k \le \frac{M\kappa}{2\varepsilon c^2}\log\left(\frac{\varepsilon_0}{\varepsilon}\right)$.

### A.2 Proof of Theorem 2

Since $\varphi_k = \theta_k - \eta\nabla\boldsymbol{\mu}^{\top}\boldsymbol{\ell}(\theta_k)$ and $\theta_{k+1} = \theta_k - \eta(\nabla\boldsymbol{\mu}^{\top}\boldsymbol{\ell}(\theta_k) + \xi_k)$, we have

$$\varphi_{k+1} = \varphi_k - \eta\xi_k - \nabla\boldsymbol{\mu}^{\top}\boldsymbol{\ell}(\varphi_k - \eta\xi_k).$$

Since the loss function $\ell_t(\theta)$ of task $t$ is $L_t$-one point strongly convex w.r.t. a given point $\theta_*$ after convolved with noise $\xi$, we have

$$\langle \nabla \mathbb{E}_\xi[\boldsymbol{\mu}^\top \boldsymbol{\ell}(\varphi - \eta\xi)], \varphi - \theta_* \rangle = \sum_{t=1}^{T} \mu_t \langle \nabla \mathbb{E}_\xi \boldsymbol{\ell}(\varphi - \eta\xi), \varphi - \theta_* \rangle$$

$$\geq \sum_{t=1}^{T} \mu_t L_t \|\varphi - \theta_*\|^2$$

$$\geq L\|\varphi - \theta_*\|^2,$$

where $L = \min_{1 \leq t \leq T}\{L_t\}$, the first inequality is due to $L_t$-one point strongly convex in Assumption 3, and the last inequality is due to the definition of $L$ and $\sum_{t=1}^{T} \mu_t = 1$. Then we can get

$$\mathbb{E}[\|\varphi_{k+1} - \theta_*\|^2] = \mathbb{E}[\|\varphi_k - \eta\xi_k - \nabla\boldsymbol{\mu}^\top \boldsymbol{\ell}(\varphi_k - \eta\xi_k) - \theta_*\|^2]$$

$$\leq \mathbb{E}[\|\varphi_k - \theta_*\|^2 + \|\eta\xi_k\|^2 + \|\nabla\boldsymbol{\mu}^\top \boldsymbol{\ell}(\varphi_k - \eta\xi_k)\|^2 - 2\langle \varphi_k - \theta_*, \eta\xi_k \rangle$$

$$- 2\langle \varphi_k - \theta_*, \nabla\boldsymbol{\mu}^\top \boldsymbol{\ell}(\varphi_k - \eta\xi_k) \rangle + 2\langle \nabla\boldsymbol{\mu}^\top \boldsymbol{\ell}(\varphi_k - \eta\xi_k), \eta\xi_k \rangle]$$

$$\leq \|\varphi_k - \theta_*\|^2 + \eta^2 r^2 + \mathbb{E}[\|\nabla\boldsymbol{\mu}^\top \boldsymbol{\ell}(\varphi_k - \eta\xi_k)\|^2] - 2\eta L\|\varphi_k - \theta_*\|^2$$

$$+ 2\mathbb{E}[\langle \nabla\boldsymbol{\mu}^\top \boldsymbol{\ell}(\varphi_k - \eta\xi_k) - \nabla\boldsymbol{\mu}^\top \boldsymbol{\ell}(\varphi_k), \eta\xi_k \rangle]$$

$$\leq (1 - 2\eta L)\|\varphi_k - \theta_*\|^2 + \eta^2 r^2 + \eta^2 \mathbb{E}[\|M(\theta_* - (\varphi_k - \eta\xi_k))\|^2] + 2\eta^3 r^2 M$$

$$\leq (1 - 2\eta L)\|\varphi_k - \theta_*\|^2 + \eta^2 r^2 + \eta^2 M^2\|\varphi_k - \theta_*\|^2 + \mathbb{E}[\langle \varphi_k - \theta_*, \eta\xi_k \rangle]$$

$$+ \eta^2 M^2 \mathbb{E}[\|\eta\xi_k\|^2] + 2\eta^3 r^2 M$$

$$\leq (1 - 2\eta L + \eta^2 M^2)\|\varphi_k - \theta_*\|^2 + \eta^2 r^2(1 + \eta M)^2,$$

where the second inequality is due to the convexity assumption and $\mathbb{E}[\xi_k] = 0$, the third and forth inequalities are due to the Lipschitz continuity in Corollary 1. We set $\rho = 2\eta L - \eta^2 M^2$ and $\beta = \eta^2 r^2(1 + \eta M)^2$. If $\rho \geq 0$, we have $\eta \leq \frac{L}{M^2}$, then we get

$$\mathbb{E}[\|\varphi_{k+1} - \theta_*\|^2] \leq (1 - \rho)\|\varphi_k - \theta_*\|^2 + \beta$$

$$\leq (1 - \rho)^k \|\varphi_0 - \theta_*\|^2 + \sum_{j=0}^{k-1}(1 - \rho)^j \beta$$

$$\leq (1 - \rho)^k \|\varphi_0 - \theta_*\|^2 + \frac{\beta}{\rho}.$$

So if $K \leq \frac{1}{\rho}\log\left(\frac{\rho\varepsilon_0}{\beta}\right)$, we have $\mathbb{E}[\|\varphi_{K+1} - \theta_*\|^2] \leq \frac{2\beta}{\rho}$. Then by the Markov inequality, with probability at least $1 - \delta$, we have

$$\|\varphi_K - \theta_*\|^2 \leq \frac{2\beta}{\rho\delta}.$$

### A.3 Noise Upper Bound

Suppose the noise produced by the EW method is $\bar{\xi} = \|\nabla\boldsymbol{\mu}^\top \boldsymbol{\ell}(\tilde{\boldsymbol{\mathcal{D}}};\theta) - \nabla\boldsymbol{\mu}^\top \boldsymbol{\ell}(\boldsymbol{\mathcal{D}};\theta)\|$ and $\|\bar{\xi}\|^2 \leq R$. The noise produced by the RLW method is $\xi = \|\nabla\boldsymbol{\lambda}^\top \boldsymbol{\ell}(\tilde{\boldsymbol{\mathcal{D}}};\theta) - \nabla\boldsymbol{\mu}^\top \boldsymbol{\ell}(\boldsymbol{\mathcal{D}};\theta)\|$. We have

$$\|\xi\|^2 = \|\nabla\boldsymbol{\lambda}^\top \boldsymbol{\ell}(\tilde{\boldsymbol{\mathcal{D}}};\theta) - \nabla\boldsymbol{\mu}^\top \boldsymbol{\ell}(\tilde{\boldsymbol{\mathcal{D}}};\theta) + \nabla\boldsymbol{\mu}^\top \boldsymbol{\ell}(\tilde{\boldsymbol{\mathcal{D}}};\theta) - \nabla\boldsymbol{\mu}^\top \boldsymbol{\ell}(\boldsymbol{\mathcal{D}};\theta)\|^2$$

$$= \|(\boldsymbol{\lambda}^\top - \boldsymbol{\mu}^\top)\nabla\boldsymbol{\ell}(\tilde{\boldsymbol{\mathcal{D}}};\theta)\|^2 + 2\left\langle (\boldsymbol{\lambda}^\top - \boldsymbol{\mu}^\top)\boldsymbol{\ell}(\tilde{\boldsymbol{\mathcal{D}}};\theta), \bar{\xi} \right\rangle + \|\bar{\xi}\|^2.$$

Because the noise $\bar{\xi}$ can be any direction, there exists a constant $s > 0$ such that $\|\bar{\xi}\|^2 = R$ and $\bar{\xi} = s(\boldsymbol{\lambda}^\top - \boldsymbol{\mu}^\top)\nabla\boldsymbol{\ell}(\tilde{\boldsymbol{\mathcal{D}}};\theta)$. Then, we have $\|\xi\|^2 \leq (1 + 2s)\|\boldsymbol{\lambda} - \boldsymbol{\mu}\|^2\|\nabla\boldsymbol{\ell}(\tilde{\boldsymbol{\mathcal{D}}};\theta)\|^2 + R$. Thus, the norm of the noise provided by the RLW method has a larger supremum than EW.

# B Proof of the Mean Value $\mathbb{E}(\lambda)$

Suppose that $\tilde{\lambda}_t(t = 1, \cdots, T)$ are independent and identically distributed (i.i.d.) random variables sampled from the Uniform or standard Normal distributions and $f$ is the softmax function. Then we have $\lambda_t = \frac{\exp(\tilde{\lambda}_t)}{\sum_{m=1}^{T} \exp(\tilde{\lambda}_m)}$ and

$$\mathbb{E}(\lambda_i) = \mathbb{E}[\exp(\tilde{\lambda}_i)]\mathbb{E}\left[\frac{1}{\sum_{m=1}^{T} \exp(\tilde{\lambda}_m)}\right] + \text{Cov}\left(\exp(\tilde{\lambda}_i), \frac{1}{\sum_{m=1}^{T} \exp(\tilde{\lambda}_m)}\right),$$

where $\text{Cov}(\cdot, \cdot)$ denotes the covariance between two random variables. Since $\{\tilde{\lambda}_t\}_{t=1}^{T}$ are i.i.d random variables, we have $\mathbb{E}[\exp(\tilde{\lambda}_i)] = \mathbb{E}[\exp(\tilde{\lambda}_j)]$ and $\text{Cov}(\exp(\tilde{\lambda}_i), 1/\sum_{m=1}^{T} \exp(\tilde{\lambda}_m)) = \text{Cov}(\exp(\tilde{\lambda}_j), 1/\sum_{m=1}^{T} \exp(\tilde{\lambda}_m))$. Therefore, we obtain

$$\mathbb{E}(\lambda_i) = \mathbb{E}(\lambda_j), \forall 1 \leq i, j \leq T.$$

Moreover, we have

$$\sum_{t=1}^{T} \mathbb{E}(\lambda_t) = \sum_{t=1}^{T} \frac{\sum_{k=1}^{K} \lambda_t^k}{K} = \frac{\sum_{k=1}^{K} \sum_{t=1}^{T} \lambda_t^k}{K} = 1.$$

Thus we have $\mathbb{E}(\boldsymbol{\lambda}) = (\frac{1}{T}, \cdots, \frac{1}{T})$. Similarly, we can prove the same result for the Bernoulli and c-Bernoulli distributions with the normalization function $f$ as $f(\tilde{\boldsymbol{\lambda}}) = \tilde{\boldsymbol{\lambda}}/(\sum_{t=1}^{T} \tilde{\lambda}_t)$.

# C Additional Experimental Results

## C.1 Results on the CityScapes Dataset

Table 6: Performance on the **CityScapes** dataset with two tasks: 7-class semantic segmentation and depth estimation. The best results for each task on each measure over loss/gradient balancing methods are marked with superscript $*$/†. The best results for each task on each measure are highlighted in **bold**. ↑ (↓) means the higher (lower) the result, the better the performance.

| | Methods | Segmentation | | Depth | | $\boldsymbol{\Delta_p}\uparrow$ |
| | | mIoU↑ | Pix Acc↑ | Abs Err↓ | Rel Err↓ | |
|---|---|---|---|---|---|---|
| | EW | 68.71 | 91.50 | 0.0132 | 45.58 | +0.00% |
| Loss Bal. | UW | 68.84 | 91.53 | 0.0132 | 46.18 | -0.09% |
| | DWA | 68.56 | 91.48 | 0.0135 | 44.49 | +0.05% |
| | IMTL-L | **69.71**$^*$ | 91.77$^*$ | 0.0128$^*$ | 45.08 | +1.58%$^*$ |
| | MOML | 69.34 | 91.65 | 0.0129 | 46.33 | +0.59% |
| | **RLW (ours)** | 68.78 | 91.45 | 0.0134 | **43.68**$^*$ | +0.69% |
| Gradient Bal. | MGDA-UB | 68.41 | 91.13 | **0.0124**$^†$ | 46.85 | +0.64% |
| | GradNorm | 68.60 | 91.48 | 0.0133 | 45.32 | +0.01% |
| | PCGrad | 68.54 | 91.47 | 0.0135 | 44.82 | -0.10% |
| | GradDrop | 68.62 | 91.45 | 0.0136 | 45.05 | -0.42% |
| | IMTL-G | 68.62 | 91.48 | 0.0130 | 44.29 | +1.09% |
| | GradVac | 68.60 | 91.47 | 0.0134 | 44.92 | -0.06% |
| | CAGrad | 68.89 | 91.50 | 0.0128 | 44.72 | +1.38% |
| | RotoGrad | 68.96 | 91.47 | 0.0127 | 43.85$^†$ | +2.13% |
| | **RGW (ours)** | 69.68$^†$ | **91.85**$^†$ | 0.0127 | 43.91 | **+2.36%**$^†$ |

**Dataset.** The CityScapes dataset (Cordts et al., 2016) is a large-scale urban street scene understanding dataset and it is comprised of a diverse set of stereo video sequences recorded from 50 different cities in fine weather during the daytime. It contains 2,975 and 500 annotated images for training and test, respectively. This dataset includes two tasks: 7-class semantic segmentation and depth estimation.

**Implementation Details.** For the CityScapes dataset, the network architecture and optimizer are the same as those in the NYUv2 dataset. We resize all the images to $128 \times 256$ and set the batch size to 64 for training. We use the cross-entropy loss and $L_1$ loss for the semantic segmentation and depth estimation tasks, respectively.

**Results.** The results on the CityScapes dataset are shown in Table 6. The empirical observations are similar to those on the NYUv2 dataset in Table 1. Firstly, both the RLW and RGW strategies significantly outperform the EW method. Secondly, the RLW method can outperform most of the loss balancing baselines except the IMTL-L method. Moreover, the RGW method achieves 2.36% performance improvement and outperforms all of the baselines.

### C.2 Results on the CelebA Dataset

**Dataset.** The CelebA dataset (Liu et al., 2015) is a large-scale face attributes dataset with 202,599 face images, each of which has 40 attribute annotations. It is split into three parts: 162,770, 19,867, and 19,962 images for training, validation, and testing, respectively. Hence, this dataset contains 40 tasks and each task is a binary classification problem for one attribute.

**Implementation Details.** We use the ResNet-18 network as a shared feature extractor and a fully connected layer with two output units as a task-specific head for each task. All the images are resized to $64 \times 64$. The Adam optimizer with the learning rate as $10^{-3}$ is used for training and the batch size is set to 512. The cross-entropy loss is used for the 40 tasks.

Table 7: Average classification accuracy (%) of different methods on the **CelebA** dataset with forty tasks. The best results over loss/gradient balancing methods are marked with superscript $*/\dagger$. The best results are highlighted in **bold**.

| | Methods | Avg Acc |
|---|---|---|
| | EW | 90.70 |
| Loss Bal. | UW | 90.84 |
| | DWA | 90.77 |
| | IMTL-L | 90.46 |
| | MOML | **90.94**$^*$ |
| | **RLW (ours)** | 90.73 |
| Gradient Bal. | MGDA-UB | 90.40 |
| | GradNorm | 90.77 |
| | PCGrad | 90.85$^\dagger$ |
| | GradDrop | 90.71 |
| | IMTL-G | 90.80 |
| | GradVac | 90.75 |
| | CAGrad | 90.72 |
| | RotoGrad | 90.45 |
| | **RGW (ours)** | 90.00 |

**Results.** Since the number of tasks in the CelebA dataset is large, we only report the average classification accuracy on the forty tasks in Table 7. According to the results, the proposed RLW strategy slightly

outperforms the EW method and performs comparably with loss balancing baseline methods. However, we can find that the RGW method and most of the gradient balancing methods are worse or achieve very limited improvement over the EW method, which indicates the gradient weighting is not suitable for the CelebA dataset.

## C.3  Results on the Office-31 and Office-Home Datasets

**Datasets.**  The Office-31 dataset (Saenko et al., 2010) consists of three domains: Amazon (**A**), DSLR (**D**), and Webcam (**W**), where each domain contains 31 object categories, and it contains 4,110 labeled images. We randomly split the whole dataset with 60% for training, 20% for validation, and the rest 20% for testing. The Office-Home dataset (Venkateswara et al., 2017) has four domains: artistic images (**Ar**), clip art (**Cl**), product images (**Pr**), and real-world images (**Rw**). It has 15,500 labeled images in total and each domain contains 65 classes. We make the same split as the Office-31 dataset. For both datasets, we consider the multi-class classification problem on each domain as a task. Similar to multilingual problems from the XTREME benchmark, each task in both Office-31 and Office-Home datasets has its own input data.

**Implementation Details.**  We use the same configuration for the Office-31 and Office-Home datasets. Specifically, the ResNet-18 network pre-trained on the ImageNet dataset is used as a shared backbone among tasks and a fully connected layer is applied as a task-specific output layer for each task. All the input images are resized to $224 \times 224$. We use the Adam optimizer with the learning rate as $10^{-4}$ and the weight decay as $10^{-5}$ and set the batch size to 128 for training. The cross-entropy loss is used for all tasks in both datasets.

**Results.**  According to the results shown in Table 8, we can see both the RLW and RGW strategies outperform the EW method on both two datasets in terms of the average classification accuracy over tasks, which implies the effectiveness of the RW methods. Moreover, the RGW method achieves the best performance (92.55% and 78.07% in term of the average accuracy) over all baselines on the Office-31 and Office-Home datasets, respectively.

Table 8: Classification accuracy (%) of different methods on the **Office-31** and **Office-Home** datasets. The best results for each domain over loss/gradient balancing methods are marked with superscript $*/\dagger$. The best results for each task are highlighted in **bold**.

| | Methods | Office-31 | | | | Office-Home | | | | |
|---|---|---|---|---|---|---|---|---|---|---|
| | | **A** | **D** | **W** | **Avg** | **Ar** | **Cl** | **Pr** | **Rw** | **Avg** |
| | EW | 82.73 | 96.72 | 96.11 | 91.85 | 62.99 | 76.48 | 88.45 | 77.72 | 76.41 |
| Loss Bal. | UW | 82.73 | 96.72* | 95.55 | 91.66 | 63.94 | 75.62 | 88.55 | 78.05 | 76.54 |
| | DWA | 82.22 | 96.72* | 96.11 | 91.68 | 63.37 | 76.05 | 89.08 | 77.62 | 76.53 |
| | IMTL-L | 83.76 | 96.72* | 95.55 | 92.01 | **65.46*** | **79.08*** | 88.45 | 78.81 | 77.95* |
| | MOML | **84.78*** | 95.08 | 96.67* | 92.17 | 64.70 | 77.03 | 88.24 | 80.00 | 77.49 |
| | **RLW (ours)** | 83.76 | 96.72* | 96.67* | 92.38* | 62.80 | 76.48 | **90.57*** | **80.21*** | 77.52 |
| Gradient Bal. | MGDA-UB | 81.02 | 95.90 | **97.77†** | 91.56 | 64.32 | 75.29 | 89.72 | 79.35 | 77.17 |
| | GradNorm | 83.93 | **97.54†** | 94.44 | 91.97 | **65.46†** | 75.29 | 88.66 | 78.91 | 77.08 |
| | PCGrad | 82.22 | 96.72 | 95.55 | 91.49 | 63.94 | 76.05 | 88.87 | 78.27 | 76.78 |
| | GradDrop | 84.27† | 95.08 | 96.11 | 91.82 | 64.70 | 77.03 | 88.02 | 79.13 | 77.22 |
| | IMTL-G | 82.22 | 95.90 | 96.11 | 91.41 | 63.37 | 76.05 | 89.19 | 79.24 | 76.96 |
| | GradVac | 82.73 | **97.54†** | 95.55 | 91.94 | 63.18 | 76.48 | 88.66 | 77.83 | 76.53 |
| | CAGrad | 82.22 | 96.72 | 96.67 | 91.87 | 63.75 | 75.94 | 89.08 | 78.27 | 76.75 |
| | RotoGrad | 82.90 | 96.72 | 96.11 | 91.91 | 61.85 | 77.03 | **90.36†** | 78.59 | 76.95 |
| | **RGW (ours)** | 84.27† | 96.72 | 96.67 | **92.55†** | 65.08 | **78.65†** | 88.66 | 79.89† | **78.07†** |

