# OpenReview forum: "Reasonable Effectiveness of Random Weighting: A Litmus Test for Multi-Task Learning"
_TMLR — Accepted by TMLR_

### Review · Reviewer_9Zr7 · 2022-08-22

**Summary Of Contributions:**

This paper proposed a novel way in understanding deep multitask learning. Specifically, rather than learning from data, the weights of learning each task are generated by a sort of **random distribution**. The authors claimed that it would help them escape the local minimum and improve prediction performance. This paper was empirically validated on a variety of real-world datasets, with the empirical performance consistently outperforming baselines.


**Requested Changes:**

See the comments and questions in
-  Are the claims made in the submission supported by accurate, convincing and clear evidence?
-  Minor points



**Strengths And Weaknesses:**

Overall, I really like the proposed idea because it is conceptually simple and appealing. Furthermore, the authors attempted to provide various theoretical and practical evidence to support the proposed approach. I also think there exist several weak points. My comments, which are based on TMLR guidelines, are as follows:

### 1. Would some individuals in TMLR's audience be interested in the findings of this paper?

Definity **YES**.

1. This paper enhances an intriguing viewpoint: even if the weight of each task is generated at random, empirical performance may be comparable to or better than other modern baselines. Because the random number generation is simple, it would be quite promising in practice.
2. Extensive empirical validations are conducted to justify the claims.
3. A convergence analysis of optimization is proposed. (However, this reviewer is concerned about the results.)

### 2. Are the claims made in the submission supported by accurate, convincing and clear evidence?

**Partially**.

The authors attempted to provide evidence from either theoretical or practical perspectives. Several arguments, however, are neither accurate nor clear. The detailed comments are as follows:


1. This paper concentrated primarily on the multi-head based deep multi-task learning problem. In other words, there is a shared parameter $\theta$ and a task-specific parameter $\phi_i$. To better fit the paper scope, I think this paper should clearly illustrate this in the introduction section.
2. [Related work] The introduction of loss balance and gradient balance approaches (in Section 2) could be improved. The related introductions/key works (such as uncertain weights) appear ambiguous, whereas I mean the related works could be listed as a table to demonstrate different techniques. Besides, it would be highly appreciated to discuss other sorts of multi-task learning perspectives such as task-similarity based MTL (e.g., paper [1])
3. In Section 3, why is further normalisation required if the task weights are generated by a distribution $p(\lambda)$?
4. Algo 1 and Algo 2 had very similar structures, and I think these two tables should be combined into a single algorithm.
5. (Major improvement) I couldn't figure out why it's **theoretically** beneficial in the theoretical analysis. (1) Theorem 1 proposed the optimization convergence behaviour on the proposed algorithm, despite the fact that the theory appears to be quite standard in the optimization. (2) Theorem 2 yielded a number of results in noisy gradient descent, where noise is injected for the gradient (this surely guarantees to escape a local minimum). This does not, in my opinion, demonstrate that noisy $\lambda$ guarantees a better generalisation property. When we look closely at the proposed framework, we can see that it is essentially a randomised version of task weights. In general, the randomised algorithm may produce more robust results with improved generalisation properties. As a result, I think Theorems 1 and 2 cannot explain the success. According to the TMLR guidelines, I would recommend revising this section to better reflect the contribution.
6. The proposed algorithm is an alternating optimization in which the task weights and model parameters are alternately updated. In fact, as hyper-parameters, $\lambda$ is essentially a function of the model parameter. As a result, I would propose a discussion about this section.
7. It is not surprising that the empirical results are nearly comparable or slightly better than the baselines. I would recommend a clear statement on the results; it is perfectly fine that the empirical performances are not always SOTA, and a clear discussion would be great.

### 3. Minor points

I would suggest that the authors carefully revise the paper because there are several typos and low-level writing issues. E.g,

- Intro “This phenomenon is related to …” should be ‘is referred to as’
- Intro ‘We think that baseline is not sufficient … to conduct a random experiment, which is missing, as a baseline to test them ’ What is a random experiment?
- The title in Sec 2 could be expressed as preliminary or problem set-up.
- Sec 2 “in every iteration” should be “in each iteration”
- Sec 3 (page 4) “,has a more stable performance.. In Sec 5.3’ I think this part is quite confusing. Sec 5.3 did not show a stable performance?
- Figure 2 is difficult to understand. I couldn't understand its main point.

Ref [1] A Principled Approach for Learning Task Similarity in Multitask Learning. IJCAI 2019.

---

> ### Author Response · Authors · 2022-09-15
> **Response to Reviewer 9Zr7 (Part 1/2)**
>
> We thank you for the detailed and constructive comments.
>
> ----
>
> **Q1: This paper concentrated primarily on the multi-head based deep multi-task learning problem. In other words, there is a shared parameter $\theta$ and a task-specific parameter $\phi_i$. To better fit the paper scope, I think this paper should clearly illustrate this in the introduction section.**
>
> A: We have clearly illustrated this in the 4th paragraph in Section 1 of the revised manuscript.
>
> ----
>
> **Q2: [Related work] The introduction of loss balance and gradient balance approaches (in Section 2) could be improved. The related introductions/key works (such as uncertain weights) appear ambiguous, whereas I mean the related works could be listed as a table to demonstrate different techniques. Besides, it would be highly appreciated to discuss other sorts of multi-task learning perspectives such as task-similarity based MTL (e.g., paper [1])**
>
> Ref [1] A Principled Approach for Learning Task Similarity in Multitask Learning. IJCAI 2019.
>
> A: We have added a more detailed introduce to related works in Section 6 of the revised manuscript.
>
> ----
>
> **Q3: In Section 3, why is further normalisation required if the task weights are generated by a distribution $p(\lambda)$?**
>
> A: As mentioned in the third paragraph of Section 3, we need to guarantee the weights to be in a simplex set (i.e., the sum of weights is one and each weight is non-negative), which is a common assumption in MTL methods.
>
> ----
>
> **Q4: Algo 1 and Algo 2 had very similar structures, and I think these two tables should be combined into a single algorithm.**
>
> A: We have combined these two algorithms into a single one, which is shown in Algorithm 1 in the revised manuscript.
>
> ----
>
> **Q5: (Major improvement) I couldn't figure out why it's theoretically beneficial in the theoretical analysis. (1) Theorem 1 proposed the optimization convergence behaviour on the proposed algorithm, despite the fact that the theory appears to be quite standard in the optimization. (2) Theorem 2 yielded a number of results in noisy gradient descent, where noise is injected for the gradient (this surely guarantees to escape a local minimum). This does not, in my opinion, demonstrate that noisy $\lambda$ guarantees a better generalisation property. When we look closely at the proposed framework, we can see that it is essentially a randomised version of task weights. In general, the randomised algorithm may produce more robust results with improved generalisation properties. As a result, I think Theorems 1 and 2 cannot explain the success. According to the TMLR guidelines, I would recommend revising this section to better reflect the contribution.**
>
> A: We have improved this part in the revision.
>
> (1) In the revised Theorem 1, we have proven the convergence property under some weak assumptions (i.e., the Polyak-Lojasiewicz condition) which can hold for deep neural networks.
>
> (2) Theorem 2 indicates that both the EW and RLW methods have high probabilities to converge to a local minimum with a radius $r$ due to the injected noise. The randomness of RLW is from both loss weights and data sampling and it can also be cast as a noise injected into the full gradient as shown in Section 4.2 of the revised manuscript. Hence, the noise in the RLW method has a larger radius $r$ than that of the EW method, making the RLW method converge to a flatter local minimum and achieve better generalization performance than the EW method, which could be verified by the newly added experiment on a synthetic dataset in Section 5.1. Moreover, we have rephrased this part (i.e., Section 4.2 in the revision) to make it more clear and easier to understand.
>
> ----
>
> **Q6: The proposed algorithm is an alternating optimization in which the task weights and model parameters are alternately updated. In fact, as hyper-parameters, $\lambda$ is essentially a function of the model parameter. As a result, I would propose a discussion about this section.**
>
> A: Indeed, from the perspective of hyper-parameter optimization, the loss weight $\lambda$ is the hyper-parameter of the MTL model and is a function of the model parameter. Actually, the MOML method, one of the baselines, has the same motivation to learn $\lambda$ in a meta-learning-based hyper-parameter optimization way. From the result in our manuscript, the proposed RLW method outperforms the MOML method in some datasets. We have added discussions in Section 6 of the revised manuscript.
>
> ----
>
> **Q7: It is not surprising that the empirical results are nearly comparable or slightly better than the baselines. I would recommend a clear statement on the results; it is perfectly fine that the empirical performances are not always SOTA, and a clear discussion would be great.**
>
> A: We have clearly illustrated this in the conclusion of the revised manuscript.

---

> ### Author Response · Authors · 2022-09-15
> **Response to Reviewer 9Zr7 (Part 2/2)**
>
> **Q8: Intro “This phenomenon is related to …” should be ‘is referred to as’**
>
> A: We have corrected it.
>
> ----
>
> **Q9: Intro ‘We think that baseline is not sufficient … to conduct a random experiment, which is missing, as a baseline to test them ’ What is a random experiment?**
>
> A: We have rephrased it in the revised manuscript.
>
> ----
>
> **Q10: The title in Sec 2 could be expressed as preliminary or problem set-up.**
>
> A: We have modified the title in Section 2 to be `Preliminary' in the revised manuscript.
>
> ----
>
> **Q11: Sec 2 “in every iteration” should be “in each iteration”**
>
> A: We have corrected it.
>
> ----
>
> **Q12: Sec 3 (page 4) “,has a more stable performance.. In Sec 5.3’ I think this part is quite confusing. Sec 5.3 did not show a stable performance?**
>
> A: This sentence and Section 5.3, which is Section 5.4 in the revised manuscript, show that by using different sampling distributions, the RLW method has different stability. According to the result in Figure 1, which is Figure 2 in the revised manuscript, the standard normal distribution has a more stable performance than others. Thus, we lean to use the standard normal distribution in the RLW method. We have clarified this sentence in the revised manuscript.
>
> ----
>
> **Q13: Figure 2 is difficult to understand. I couldn't understand its main point.**
>
> A: Figure 2, which is Figure 3 in the revised manuscript, shows the convergence speed of EW and RLW methods on different datasets with different numbers of tasks. This figure shows that RLW with standard normal distribution converges as fast as EW, while the convergence speed with the c-Bernoulli distribution becomes slower when the number of tasks increases. Thus, we lean to use the standard normal distribution in RLW method. We have further clarified this in the caption of Figure 3 in the revision.

---

> ### Comment · Reviewer_9Zr7 · 2022-09-15
> **Post-rebuttal**
>
> Dear authors,
>
> Thanks for your detailed responses. For me, the technical concerns are all properly addressed. Thus I would definitely support the acceptance.
>
> As for the paper writing, I quickly went through the paper and found it is much better now. While I occasionally found several low-level writing issues, thus I would suggest maybe a refinement.

---

> > ### Author Response · Authors · 2022-10-16
> > **Thank you!**
> >
> > We are glad that our responses help you better understand our method. We will refine the paper writing better.

---

### Review · Reviewer_KBdf · 2022-08-24

**Summary Of Contributions:**

This paper proposes to explore the use of random weights in multi-task learning (MTL). The main underlying idea is to study the performance of using random weights drawn from some probability distribution either when reweighing the individual tasks loss terms optimized by MTL algorithm or when defining the weights of individual tasks’ gradients. The authors first study theoretically their proposed reweighing strategy and show that it 1) converges to an optimal solution within a certain radius and 2) may avoid bad local minima. Then, they present an experimental study revealing that such a simple method can represent a surprisingly strong baseline that remains on par or better than other common weighing strategies and is always better than equal weighing.

I believe that this paper presents a strong contribution to the multi-task learning community by studying a very intuitive, yet overlooked baseline.


**Requested Changes:**

Major:

My major requested change is regarding the theoretical part where the authors do not clearly state to which extent the assumptions made are restrictive. Below, I explain this in more detail:
1. It seems that assuming strong convexity of the loss with respect to $\theta$ in Theorem 1 means that the authors do not consider neural networks which optimize a function that is non-convex in its parameters. It is not stated explicitly but that would limit the scope of this result to some extent as most common MTL methods are based on deep neural networks.

2. In Theorem 2, the authors introduce the notion of local minimum convoluted with noise even though it was never mentioned before. Theorem 2 in general lacks a proper introduction. Ideally, the authors would spend more time to first introducing the intuition and the motivation to have the quantities that they have in the statement. Currently, they are introduced directly in the statement which is cumbersome and makes the whole results less clear.

3. It is a good practice also to explain the pitfall scenarios for random weight generation. Currently, one has an impression that this baseline works well in all cases. Did the authors identify the scenarios where random weights are worse than equal ones?

Minor changes:

Some typos:
1. “Task balancing” in the abstract is not introduced, consider rephrasing without this specific term
2. “thus it is apparently” -> “it is apparent”
3. “RLW has a better generalization performance that EW” -> than
4. “bring negligibly additional computational costs” -> negligible


**Strengths And Weaknesses:**

Strengths:
1. This paper is easy to read and it is concise while being mostly self-contained. For the most part, the reader not familiar with current multi-task learning state-of-the-art techniques for loss/gradient weighting should be able to get a basic understanding of the methods from reading the first sections of this paper.
2. The theoretical analysis is a nice backup for the experimental proposal.
3. Experimental study presents results across a wide range of benchmarks with different tasks and of different modalities.

Weaknesses:
1. The assumptions used to prove Theorems 1 and 2 should be better clarified. Ideally, it would be nice to separate the assumptions from the statements of these two theorems into a bullet list and explain each assumption individually.
2. I had a feeling that this paper was written in a hurry as it has a fair amount of typos and grammatical mistakes. Major proofreading would be needed to fix those.

---

> ### Author Response · Authors · 2022-09-15
> **Response to Reviewer KBdf**
>
> We thank you for the detailed and constructive comments.
>
> ---
>
> **Q1: It seems that assuming strong convexity of the loss with respect to $\theta$ in Theorem 1 means that the authors do not consider neural networks which optimize a function that is non-convex in its parameters. It is not stated explicitly but that would limit the scope of this result to some extent as most common MTL methods are based on deep neural networks.**
>
> A: In the revised Theorem 1, we have proven the convergence property under some weak assumptions (i.e., the Polyak-Lojasiewicz condition) which can be held for deep neural networks. Please refer to Section 4.1 in the revised manuscript.
>
> ---
>
> **Q2: In Theorem 2, the authors introduce the notion of local minimum convoluted with noise even though it was never mentioned before. Theorem 2 in general lacks a proper introduction. Ideally, the authors would spend more time to first introducing the intuition and the motivation to have the quantities that they have in the statement. Currently, they are introduced directly in the statement which is cumbersome and makes the whole results less clear.**
>
> A: We have revised this part to make it more clear and easier to understand. Please refer to Section 4.2 in the revised manuscript.
>
> ---
>
> **Q3: It is a good practice also to explain the pitfall scenarios for random weight generation. Currently, one has an impression that this baseline works well in all cases. Did the authors identify the scenarios where random weights are worse than equal ones?**
>
> A: Extensive experiments conducted in synthetic and real-world datasets consistently show the RW methods can outperform the EW method. We will continue to study it and identify conditions where RW performs worse than EW.
>
> ---
>
> **Q4: “Task balancing” in the abstract is not introduced, consider rephrasing without this specific term.**
>
> A: We have rephrased it in the revised manuscript.
>
> ---
>
> **Q5: “thus it is apparently” $\rightarrow$ “it is apparent”**
>
> A: We have corrected this typo.
>
> ---
>
> **Q6: “RLW has a better generalization performance that EW” $\rightarrow$ than**
>
> A: We have corrected this typo.
>
> ---
>
> **Q7: “bring negligibly additional computational costs” $\rightarrow$ negligible**
>
> A: We have corrected this typo.

---

> ### Author Response · Authors · 2022-10-16
> **Have our responses addressed your concerns?**
>
> Dear Reviewer KBdf,
>
> We hope the responses and the revised manuscript can address your concerns and eagerly look forward to receiving your feedback.
>
> Thank you!

---

### Review · Reviewer_yYkZ · 2022-09-03

**Summary Of Contributions:**

The paper considers the specification of task weights in multi-task learning. Specifically, two types of weighting are considered: loss weights which affect shared and task parameters and gradient weights that affect only the shared parameter. For these types, the paper proposes random weightings as competitors which should be used in substantiating performance of multi-task learning algorithms that focus on weighting schemes.

As evidence that these random weighting schemes should be used, the paper supplies analyses proving convergence of the shared task parameter under random loss weights in expectation and with high probability (the latter with additional assumptions). The paper also supplies experiments showing random weightings are at least competitive with the standard baseline of equal weighting if not often-times superior in performance in a computer vision dataset and NLP dataset and are robust to choice of randomness and NN architecture.

I think this is a very good paper with a nice mix of theory and experiments. The proposal to use random weightings as a standard competitor in future MTL algorithm performance claims is quite achievable and convincingly backed-up. (Note, I did not check the proofs in great detail beyond general plausibility.)

**Requested Changes:**

I have no requested changes.

**Strengths And Weaknesses:**

Strengths:

* Proposal to utilize random weightings as additional baselines for future MTL algorithms is very low cost to implement and should be easily achievable.

* Combination of analyses and experiments provides compelling evidence to support claim.

* The paper is clear and organized well.

* Experiments are performed wrt/ a variety of performance dimensions: application, robustness, speed, experimental (combining loss and gradient weightings)

Weaknesses:

* The theory and experiments are a little too complementary in that they are almost orthogonal. In other words, it would have been nice if some synthetic experiment(s) was provided showing the analytical claims held. However, I consider this absence very minor in the overall amount of evidence supplied.

---

> ### Author Response · Authors · 2022-09-15
> **Response to Reviewer yYkZ**
>
> We thank you for the detailed and constructive comments.
>
> ----
>
> **Q1: The theory and experiments are a little too complementary in that they are almost orthogonal. In other words, it would have been nice if some synthetic experiment(s) was provided showing the analytical claims held. However, I consider this absence very minor in the overall amount of evidence supplied.**
>
> A: We have added experiments on a toy problem in Section 5.1 to empirically show that the proposed RW method can converge to a flatter minimum than the EW method, which could verify the theoretical results.

---

> ### Author Response · Authors · 2022-10-16
> **Have our responses addressed your concerns?**
>
> Dear Reviewer yYkZ,
>
> We hope the responses and the revised manuscript can address your concerns and eagerly look forward to receiving your feedback.
>
> Thank you!

---

### Author Response · Authors · 2022-09-15
**Summary of Change**

We thank the action editor and all of the reviewers for their time and effort in providing insightful comments for our manuscript. Based on these reviews, we have revised our manuscript and uploaded the new version. All the changes are highlighted in **blue** for quick identification in the revised manuscript. For each reviewer, we have also made a detailed and individual response and we will be happy to further discussions if any part remains unclear.

To summarise, we have made the following changes to the manuscript:

- We have combined Algorithms 1 and 2 into a single table to make it more clear and more concise.

- For the theoretical analysis in Section 4, we have proven Theorem 1 under some weak assumptions which can be held for non-convex problems, and rephrased Theorem 2 to make it more clear and easier to understand.

- We have added a toy example in Section 5.1 to empirically show the proposed RW method can converge to a flatter minimum than the EW method.

- We have added Section 6 to overview related works.

- We have corrected some typos.

---

### Decision · Action_Editors · 2022-10-19

**Recommendation:** Accept with minor revision

**Comment:**

The paper examines a simple yet effective method for multi-task learning. The method weights each task randomly through its loss or gradient. The authors demonstrate that random weighting improves generalization performance, allowing the method to perform competitively to other state-of-the-art methods. Some theoretical analyses on the convergence property of the method are provided to strengthen the paper.

The reviewers appreciate the simplicity of the proposed method that comes with competitive performance to serve as a solid future baseline for multi-task learning. The breadth of the experiments is sufficient to demonstrate the strength of the proposed method. Before the revision, most concerns surrounded how the theory connected to the experiments. The authors attempted to resolve the concerns by providing more toy experiments, clarifying and loosening the theory's assumptions, and rephrasing several hand-waving claims. The efforts convinced the reviewers. Ultimately, all reviewers unanimously agreed that the work could be accepted.

Some reviewers are still aware of typos and grammatical errors in the latest version. The authors should do careful proofreading when preparing for the camera-ready version.


**Audience:**

Surely!

**Claims And Evidence:**

Yes, the reviewers agree that the claims made in the submission supported by accurate, convincing and clear evidence.

---

> ### Author Response · Authors · 2022-10-23
> **Camera-ready Version**
>
> Dear Action Editors,
>
> Thanks for your and all reviewers' effort. We have updated the camera-ready version of our paper.